# Manipulating the diffusion energy barrier at the lithium metal electrolyte interface for dendrite-free long-life batteries

Jyotshna Pokharel[1,2], Arthur Cresce [3], Bharat Pant [4], Moon Young Yang [5], Ashim Gurung[2], Wei He[2], Abiral Baniya[2], Buddhi Sagar Lamsal[2], Zhongjiu Yang [1], Stephen Gent[6], Xiaojun Xian [2], Ye Cao [4] ✉, William A. Goddard III [5] ✉, Kang Xu [3,7] ✉ & Yue Zhou [1] ✉

Constructing an artificial solid electrolyte interphase (SEI) on lithium metal electrodes is a promising approach to address the rampant growth of dangerous lithium morphologies (dendritic and dead $Li^0$) and low Coulombic efficiency that plague development of lithium metal batteries, but how $Li^+$ transport behavior in the SEI is coupled with mechanical properties remains unknown. We demonstrate here a facile and scalable solution-processed approach to form a $Li_3N$-rich SEI with a phase-pure crystalline structure that minimizes the diffusion energy barrier of $Li^+$ across the SEI. Compared with a polycrystalline $Li_3N$ SEI obtained from conventional practice, the phase-pure/single crystalline $Li_3N$-rich SEI constitutes an interphase of high mechanical strength and low $Li^+$ diffusion barrier. We elucidate the correlation among $Li^+$ transference number, diffusion behavior, concentration gradient, and the stability of the lithium metal electrode by integrating phase field simulations with experiments. We demonstrate improved reversibility and charge/discharge cycling behaviors for both symmetric cells and full lithium-metal batteries constructed with this $Li_3N$-rich SEI. These studies may cast new insight into the design and engineering of an ideal artificial SEI for stable and high-performance lithium metal batteries.

The increasing demand for rechargeable energy sources to power electronics, electric vehicles, and large-scale grid energy storage has driven extensive research of energy-dense lithium-based batteries[1–3]. To meet such demand, high energy density batteries other than state-of-the-art lithium-ion batteries (LIBs) with typical specific energies above 300 Wh kg$^{-1}$ must be developed[4,5]. The metallic lithium negative electrode has a high theoretical specific capacity (3857 mAh g$^{-1}$) and a low reduction potential (−3.04 V vs standard hydrogen electrode),

making it the ultimate choice of negative electrode material for high energy Li-based rechargeable batteries[1,6–8]. Although Li metal ($Li^0$) negative electrodes potentially enable batteries with high energy density, they tend to form dangerous $Li^0$ morphologies (dendritic and the subsequent mossy $Li^0$) and induce sustained electrolyte decomposition, which eventually leads to poor reversibility as indicated by low Coulombic efficiency (CE) and even safety hazards[9]. Considerable efforts have been devoted to addressing the challenges of $Li^0$ negative

[1]Department of Mechanical Engineering, The University of Texas at Dallas, Richardson, TX, USA. [2]Department of Electrical Engineering and Computer Science, South Dakota State University, Brookings, SD, USA. [3]Battery Science Branch, Energy Science Division, U.S. CCDC Army Research Laboratory, Adelphi, MD, USA. [4]Department of Materials Science and Engineering, University of Texas at Arlington, Arlington, TX, USA. [5]Materials and Process Simulation Center, California Institute of Technology, Pasadena, CA, USA. [6]Department of Mechanical Engineering, South Dakota State University, Brookings, SD, USA. [7]SolidEnergy Systems LLC, 35 Cabot Rd., Woburn, MA, USA. ✉e-mail: ye.cao@uta.edu; wag@caltech.edu; kang.xu@ses.ai; zhou@utdallas.edu

electrodes, including interfacial engineering[8,10–15], electrolyte engineering[5,16–21], minimizing volume change by architecting stable hosts[22–25], and preventing dendrite propagation with modified separators[26–28].

Partial physical suppression of dendrite growth has been well achieved in the previous work, but to fully eliminate the unstable interface between the Li[0] electrode and electrolyte, one must understand the fundamental mechanism of dendrite growth. The formation of Li[0] dendrites is induced by the chemical and morphological inhomogeneity of the in-situ SEI on the Li[0] surface, which leads to uneven local current density. Efforts have been made to relate dendritic Li[0] formation to the interfacial kinetics as described by Sand's equation (Supplementary Note 1)[29,30] or to the diffusion-limited aggregation model by Chazalviel[31], but the actual factors involved are far more complicated than the diffusion models derived for metal ion deposition from aqueous electrolytes, where the interphase does not exist. Nevertheless, a non-quantitative approximate relation exists between the fractal deposition pattern and the maximum interfacial current, where rapid consumption of Li[+] in certain locations does result in concentration polarization, which invites local enrichment of Li[+] and subsequent preferential deposition[32]. Hence, the diffusion energy barrier of Li[+] at the interface should play a critical role, and various methods have been proposed to reduce the diffusion barrier[33–39]. The Li[+] transference number, which quantifies the pure contribution from Li[+] to the entire migration flux, has a decisive impact on the manner in which Li[+] approaches the interphase-enclosed Li[0] surface[40]. Consequently, the most directive and effective approach to guide Li[0] deposition in an even and homogenous manner so that dendrite formation is minimized is to decrease the diffusion energy barrier and increase the Li[+] transference number. Most current research toward increasing transference number focuses on single-ion conducting solid polymers, ceramic solid electrolytes, and their composites[41,42]. The former (polymers) offer both a rigid framework of interconnected nanopores and a high transference number, but are limited by low mechanical strength and poor ionic conductivity[41,42], while the latter (ceramics) always encounter poor contact issues of solid-solid interfacing, where the advantage of high Li[+] transference number become meaningless unless under high pressure. The brittle nature of the ceramic solid electrolytes further complicates the challenges[41–43]. Hence, it is highly desired to develop a new strategy to engineer the interface between the Li[0] electrode and the electrolyte with both near-unity Li[+] transference and robust mechanical strength to enable "dual protection" for the stabilization of the Li metal electrode.

Herein, we propose a rational design of an artificial SEI produced by treating Li[0] with tetramethylethylenediamine (TEMED), which exhibits a low diffusion energy barrier, high Li[+] transference number, and unrivaled mechanical strength to simultaneously overcome diffusion and advection-limited ion transport to achieve dendrite-free Li plating/stripping. Notably, TEMED spontaneously reacts with Li[0] upon contact and forms pure α-phase Li$_3$N. Differing from conventional Li$_3$N artificial SEI that is fabricated from the exposure of Li[0] in the N$_2$ atmosphere, an artificial SEI achieved in this way offers excellent Li[+] conductivity with a lower energy barrier for Li[+] migration, directly benefitting ion transport at the interface between electrode and electrolyte. This effectively eliminates the uneven current distribution across the electrode surface, preventing preferential local growth of Li[0] seedlings. The high modulus of Li$_3$N ensures excellent mechanical strength that tolerates volume change to enforce a more uniform Li[+] ion flux. The TEMED-treated symmetrical cell shows outstanding plating/stripping cycles with reduced overpotential and the full cell exhibits remarkably improved cycling stability and capacity retention as well as capacity utilization at high rates compared to untreated Li[0]. In this work, we demonstrate phase-pure artificial SEI on Li[0] negative electrode that is capable of resolving compounded challenges faced by Li[0] electrodes.

## Results

Lithium chips were completely immersed into TEMED in a petri dish to ensure complete passivation of Li[0] (Fig. 1a). A color change from shiny silver to light black and later to dark black is observed with the reaction time. To obtain the optimum reaction time, lithium chips were kept in the TEMED for 6 h, 12 h, and 18 h, respectively. Figure 1c shows that with a reaction time of 6 h, the film formed from TEMED has not fully covered the Li[0] surface yet. With 12 and 18 h, full coverage by the artificial SEI layer is observed. Both visual and scanning electron microscope (SEM) inspection revealed that the artificial TEMED-based SEI layer formed during 6 h of reaction time does not cover the surface completely, which does not prevent the reaction between electrolyte with Li[0], leading to consumption of both electrolyte and Li[0] resulting in low CE and capacity decay. In comparison, the artificial SEI formed during 12 or 18 h fully covers the Li[0] surface based on the SEM images, thereby preventing direct contact of the electrolyte with Li[0]. Cross-sectional SEM images (Fig. 1j–m) show the average thickness (t) of the artificial SEI layer obtained with different TEMED treatment times to be 5, 10, and 20 μm for 6, 12, and 18 reaction hours, respectively. It is assumed that a thicker SEI will have a higher Li[+] barrier energy and higher impedance, resulting in slower Li[+] diffusion. Thus, the SEI layer thickness should be optimized in order to prevent direct contact between Li[0] and electrolyte while maintaining usefully high Li[+] ion conductivity.

## Material characterization, electrochemical spectroscopy, and transference number

Contact angle measurements have been performed to determine the wettability of the electrolyte on untreated Li[0] and TEMED-treated Li[0] (Fig. 2a, b). To ensure good Li[+] ion conductivity, rate capability, and formation of a stable SEI, the electrolyte must be able to significantly wet the electrode[44]. The contact angle for untreated Li[0] is 41° suggesting poor wettability with the electrolyte which could cause lower ionic conductivity. In contrast, TEMED-treated Li[0] shows a significantly lower contact angle of 12° with electrolyte, suggesting that the TEMED-originated SEI is better interfaced with the electrolyte, which is beneficial to homogenize ion distributions in the vicinity of the negative electrode, not only supporting efficient Li[+] transport but also prevents the unwanted morphologies (dendritic or dead Li[0]) that are known to be induced by uneven Li[+] flux distribution[45]. The phase purity of TEMED-treated Li[0] was characterized by X-ray diffraction (XRD) (Fig. 2c). Distinct Li[0] peaks of (110), (200), and (211) were observed at 36°, 52°, and 65°[10] and Li$_3$N peaks of (001) and (002) at 22.96° and 46.6°, respectively. The formation of hexagonal α-phase single crystal Li$_3$N from TEMED-treated Li was also indicated by the transmission electron microscope (TEM) characterization as shown in the Supplementary Fig. 1. In comparison, conventional methods to obtain Li$_3$N, where Li[0] was treated with nitrogen flow show a polycrystalline structure with (001), (100), (002), (110), and (102) peaks for Li$_3$N (Supplementary Fig. 2). XRD of Li$_3$N after 100 cycles was also performed to examine the change of phase purity (Supplementary Fig. 3), revealing surprising structural stability of Li$_3$N formed by TEMED against charge/discharge cycling, as evidenced by the strong α-phase peaks at 22.9° (001) and 46.6° (002). These α-phase Li$_3$N diffraction peaks also reveal that the Li$_3$N film obtained from TEMED treatment is highly orientated along the direction vertical to the Li[0] surface, hence offers excellent Li[+] conductivity and implies a lower Li[+] migration energy barrier[46].

Electrochemical Impedance Spectroscopy (EIS) measurements were performed to further characterize the electrode interface. Figure 2d and Supplementary Fig. 4 compare EIS for untreated Li[0] and for TEMED-treated Li[0]. The first semicircle in the higher frequency range indicates the interfacial resistance of the artificial SEI or resistance of Li[+] flux through an artificial SEI, while the second semicircle in the lower frequency range indicates the charge transfer resistance R$_{ct}$. The

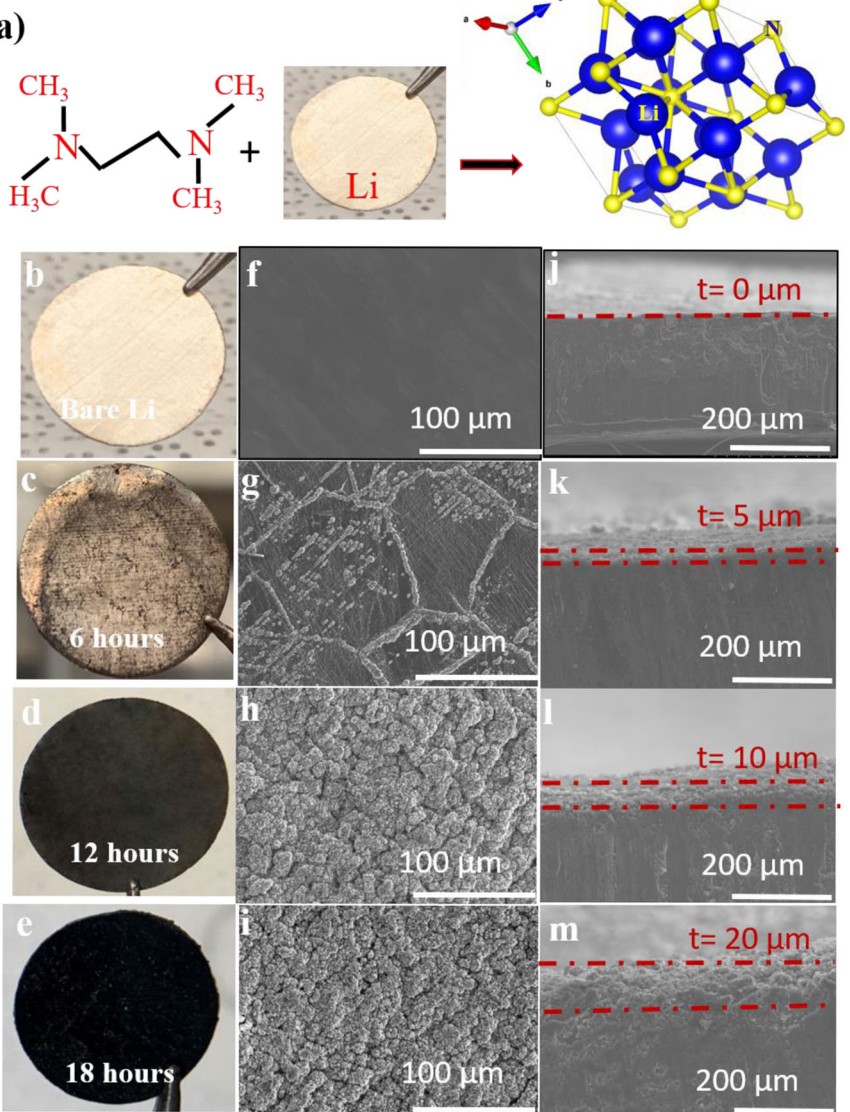

**Fig. 1 | Process of the artificial SEI layer. a** Schematic for reaction of TEMED with Li⁰ to produce lithium nitride. **b**–**e** Photographic images of bare Li⁰ and TEMED-treated Li⁰ for different treatment times. **f**–**i** The corresponding cross-sectional SEM images of bare Li⁰ and TEMED-treated Li⁰. **j**–**m** The corresponding top-view SEM images of bare Li⁰ and TEMED-treated Li⁰.

symmetrical cell constructed on untreated Li⁰ showed a high resistance of ~400 ohms, whereas for the TEMED-treated Li⁰ symmetric cell, we observed a reduced resistance of ~200 ohms. We attribute the smaller overall impedance of the TEMED-treated cell to the better Li⁺ transport performance across the Li₃N artificial SEI formed at the interface. The Li⁺ conductivity of the TEMED treated Li was calculated to be ~$4.19 \times 10^{-1}$ mS cm⁻¹ based on the series resistance of the symmetric cell, a sufficiently high value to establish a fast Li⁺ exchange channel between Li⁰ metal and electrolyte[10].

The transference number of Li⁺ at the interface between the Li⁰ electrode and the electrolyte was evaluated using Bruce-Vincent Approach. High cation transference numbers are desirable to avoid concentration gradients in the cell and to delay the nucleation and growth of lithium metal dendrites while charging the cell at a high rate. It should be noted that in conventional carbonate-based electrolytes, the transference number $t^+$ is typically between 0.1 and 0.4[47]. Although higher $t^+$ can be obtained in polymeric, ceramic, or nanoparticle-based electrolytes, in which the anions are immobilized to a stationary or slow-moving support, low ionic conductivity and other compromises in properties such as interfacing or mechanical strength always

accompany them. We expect that the lithium nitride layer with a pure α-phase on top of Li⁰ will address these conflicts. The transference numbers are determined in symmetric cells consisting of untreated Li-Li and treated Li-Li, respectively. The cell was initially conditioned to establish a stable interface by charging and discharging at 0.01 mA cm⁻², with 4-h charge, 30-min rest, and 4-h discharge, with the process repeated 6 times. The cell was then polarized at 10 mV for 10 h to ensure a steady state (Fig. 2e, f). EIS spectra before polarization and after the steady state had been reached are shown in inset of Fig. 2e, f. The steady-state cation transference number was then calculated via Eq. (1).

$$t^+ = \frac{I_s(\Delta V - I_o R_o)}{I_o(\Delta V - I_s R_s)} \tag{1}$$

where $t^+$ is the steady-state cation transference number, $\Delta V$ is the applied voltage, $I_o$ is the initial current, $I_S$ is the steady-state current, $R_o$ is the initial interfacial resistance, $R_S$ and is the steady-state interfacial resistance. The calculated result shows that the TEMED-treated Li electrode exhibits a high steady-state cation transference number of $t^+$

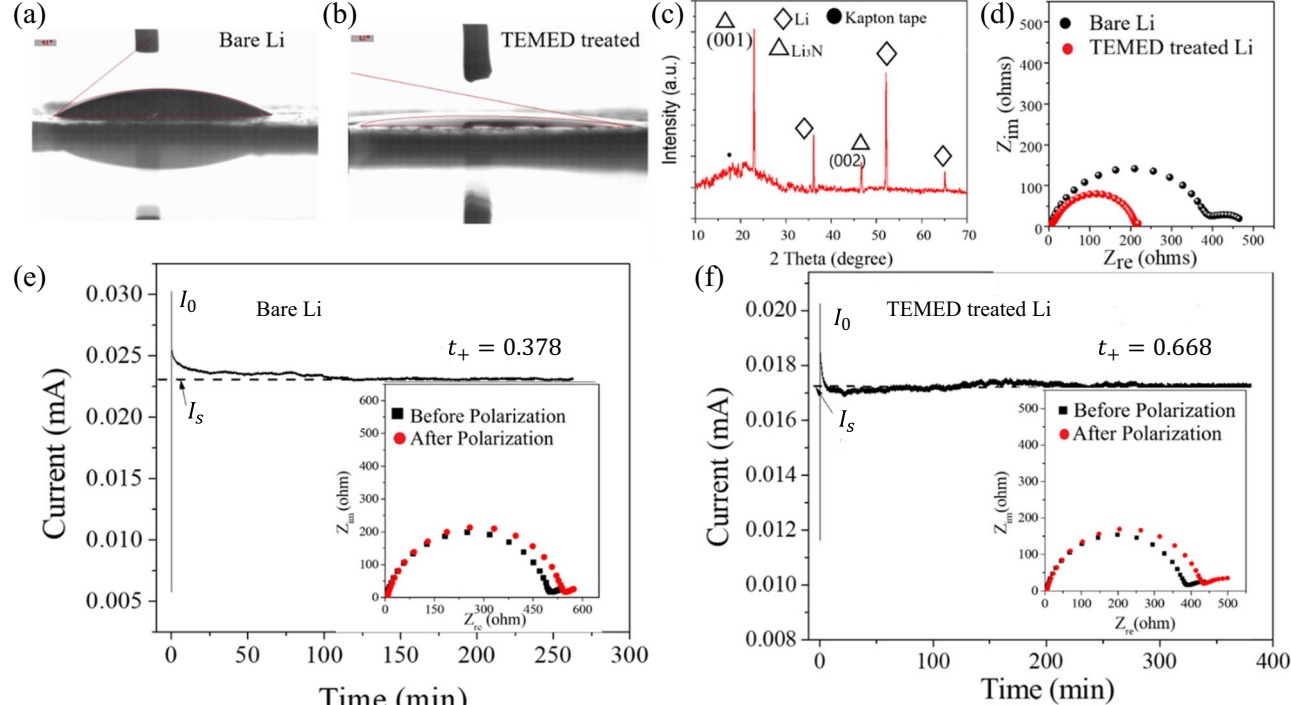

**Fig. 2 | Property comparison between untreated Li⁰ and TEMED-treated Li⁰.** Contact angle measurements of **a** untreated Li⁰ and **b** TEMED-treated Li⁰. **c** XRD spectrum of TEMED-treated Li⁰. **d** EIS measurements of untreated Li⁰ and TEMED-treated Li⁰. Steady-state current under 10 mV polarization for **e** Li-Li symmetric cell **f** TEMED-Li/TEMED-Li symmetric cell. Inset shows EIS measurements before and after polarization.

$= 0.668$, untreated in comparison with $t^+ = 0.37$ for untreated Li⁰. This result strengthens our understanding that the TEMED-treated Li⁰/electrolyte interphase plays a dominant role in altering Li⁺ transport behavior. The improvement in the cation transference number with the artificial SEI layer has the potential to suppress dendrite growth by lowering the diffusion energy barrier and regulating the ion concentration at the interface in organic electrolyte rather than solid-state electrolyte.

## Phase field simulation

The Arrhenius equation (Supplementary Note 3) indicates that a decrease in the activation energy leads to an increase in the diffusion coefficient. This decrease in activation energy for diffusion lowers Li⁺ migration energy barriers which increases ion transport at the interface between the electrode and electrolyte. The transference number is directly proportional to its diffusion coefficient. Comparative Arrhenius-plots for TEMED-treated Li⁰, N₂ treated Li⁰ and untreated Li⁰ are shown in Supplementary Fig. 5, revealing an activation energy of 0.703 eV for untreated Li⁰, 0.613 eV for N₂-treated Li⁰, and 0.48 eV. For TEMED-treated Li⁰, respectively. This successive decrease in activation energy results in a much higher Li⁺ mobility, which in turn decreases the concentration gradient across the corresponding SEI to provide a more uniform surface for Li⁺ migration and plating. To test this hypothesis and to further understand the mechanism for suppression of dendritic and dead Li⁰ growth by the Li₃N-based SEI, we further characterized the activation energy of Li⁺ using integrated phase field simulations to elucidate the fundamental correlation between our novel artificial layer with phase purity and the Li⁺ transport behavior at the interface, which we then verify with the experimental results. A highly diffusive SEI is introduced on the Li⁰ surface to mimic the treated Li⁰ covered under artificial SEI. A small protrude is introduced on the surface of the Li metal to mimic the nucleus of Li⁰. The diffusivity of Li⁺ in the electrode ($D_e$) and the electrolyte ($D_s$) are set to be $4.6 \times 10^{-13}$ cm²/s and $4.6 \times 10^{-10}$ cm²/s, respectively, while the diffusivity

in the artificial SEI layer ($D_i$) is 3 times larger than $D_s$ for N₂-treated Li⁰ and 10 times for TEMED-treated Li⁰. These values are calculated based on the activation energy obtained from experiments (Supplementary Fig. 5). Figure 3a–c shows snapshots of the Li⁰ dendrite structure on untreated Li⁰, N₂-treated Li⁰, and TEMED-treated Li⁰ having Li₃N as an artificial SEI after 400 s, respectively. For untreated Li⁰, we observe that an initial Li⁰ protrude grows into a filament-like dendritic morphology with side branches budding from the primary arm of the dendrite (Fig. 3a). For N₂-treated Li⁰ negative electrode, Li⁰ dendrite forms and grows at a smaller growth rate, and the side growth of the primary arm of Li⁰ dendrite is hardly seen (Fig. 3b). Instead, the initial Li⁰ protrude forms a dome-like morphology with a smooth electrode-electrolyte interface, and its growth rate is significantly reduced. It can thus be inferred that the artificial SEI layer of higher Li⁺ diffusivity and higher Li⁺ transference number can indeed significantly suppress the dendritic Li⁰ growth. To further elucidate our findings, we plotted the 1D evolutions of Li⁺ concentration along the x direction across the tip of the dendrite, as indicated by the arrows in the 2D inset plots (Fig. 3c, d). The Li⁺ concentration at the tip of the dendrite increases sharply for untreated Li⁰ surface, whereas it increases rather gradually for treated Li⁰.

The electric field variation ($E_x$) along the dendrite tip at different time steps for untreated Li⁰, N₂-treated, and TEMED-treated Li⁰ are also compared in Fig. 3g, i. The local electric field remains almost constant in the electrode and electrolyte, but it is maximized at the tip of the dendrite for all the cases. However, for untreated Li⁰ (Fig. 3g), the maximum $E_x$ is 2 times higher for that of the N₂-treated Li⁰ and 3 times for TEMED-treated Li⁰ (Fig. 3h, i), which is due to the sharper tip morphology with larger curvature, leading to a higher Li⁺ concentration gradient near the tip due to the higher local electric field that further facilitates the growth of the dendrite on an untreated Li⁰ surface. These results indicate that Li⁰ dendrite growth is a self-accelerating process, agreeing with previous reports[48]. We also observe that for the untreated Li⁰ $E_x$ at the dendrite tip and in the

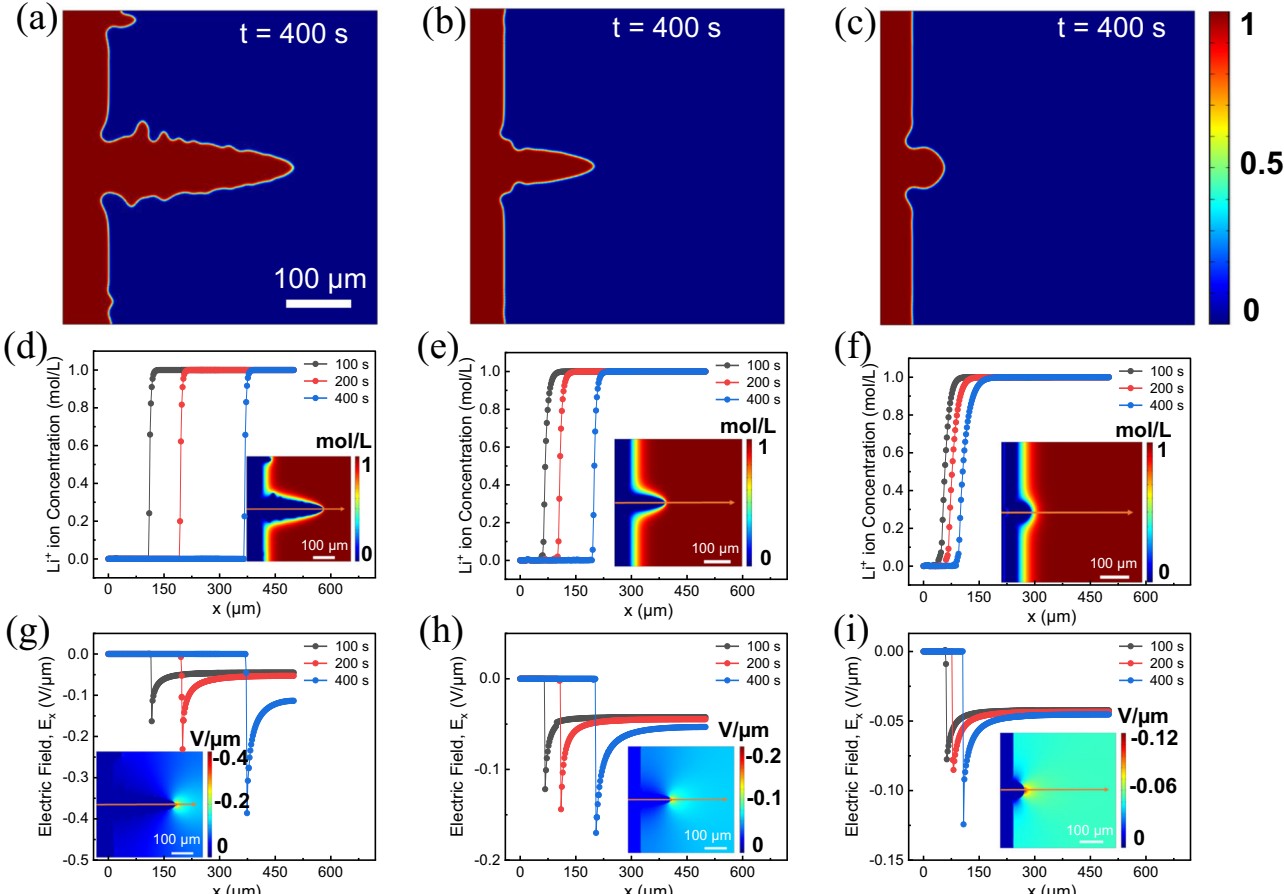

**Fig. 3 | Phase-field simulation of Li⁰ dendrite growth from untreated Li negative electrode, N₂-treated Li negative electrode, and TEMED-treated Li negative electrode covered with Li₃N SEI.** Dendrite morphology grown on **a** untreated lithium metal, **b** N₂ treated Li negative electrode, and **c** TEMED treated Li⁰ negative electrode represented by phase-field variable $\xi$. **d**–**f** 1D evolutions of Li⁺ concentration profile along x-axis across the tip of the dendrite for **d** untreated Li⁰, **e** N₂ treated Li⁰, and **f** TEMED-treated Li⁰. The images on the inset show the 2D map of the Li⁺ concentration at t = 400 s. **g**–**i** 1D evolutions of electric field profile along x-axis across the tip of dendrite for **g** untreated lithium, **h** N₂-treated Li⁰, and **i** TEMED-treated Li⁰. The images on the inset show the 2D distribution of the local electric field at t = 400 s.

electrolyte solution increases to maximum over time. Whereas, for the treated Li⁰ negative electrode, the variation in $E_x$ at different time steps is much less significant, indicating that Li⁰ dendrite growth is significantly inhibited.

### Elemental analysis, topography, and modulus mapping

X-ray photoelectron spectroscopy (XPS) was conducted on both TEMED-treated and untreated Li⁰ to decipher the chemistry of the artificial SEI as shown in Fig. 4. All the high-resolution spectrums were fitted by the Lorentzian in terms of spin-orbit doublets. Figure 4b shows the high-resolution N *1s* spectrum, with peaks at 398.3 eV assigned to Li₃N, which is known to be a good Li⁺ conductor. The absence of any N *1s* peak for untreated Li⁰ (Fig. 4e) confirms that the presence of Li₃N arises solely from the reaction between TEMED and Li⁰. The C *1s* (Supplementary Fig. 6) confirms that there is no presence of any other organic moieties, and the excellent performance of the battery is solely by the presence of Li₃N. Furthermore, we also detected the presence of N from the energy dispersive spectrum (EDS), which shows the distinct presence of N and a uniform distribution of N over the surface of the TEMED-treated Li⁰ (Supplementary Fig. 7). Based on the above analysis derived from diversified techniques, we believe that this N-rich SEI stabilizes the Li⁰/electrolyte interface, leading to uniform Li⁰ electroplating and increased cycle life.

Atomic force microscopy (AFM) was employed to visualize surface topography and measure the corresponding Young's modulus of untreated and TEMED-treated Li⁰ (Fig. 4g–j). The surface roughness values of the untreated and TEMED-treated Li⁰ were compared by measuring the average surface root mean square (RMS) via high-resolution AFM, which for untreated Li (Fig. 4g) and TEMED-treated Li⁰ (Fig. 4i) are 242 and 157 nm, respectively. The higher RMS value for untreated Li⁰ implies uneven and rough surfaces that can induce high local current at the protuberances and encourage Li⁰ dendrite on electrode surface[49,50]. In contrast, the smooth surface of TEMED-treated Li⁰ provides a route for uniform Li⁰ plating. The corresponding Young's modulus mapping values of untreated (Fig. 4h) and TEMED-treated Li (Fig. 4j) exhibit an average Young's modulus values of 0.32 and 6.85 GPa, respectively. We attribute the 20 times higher Young's modulus value of TEMED-treated Li⁰ to the superior structural efficiency and strong mechanical strength of the highly oriented α-phase Li₃N. This Young's modulus value is significantly higher than the threshold value of 6.0 GPa for Li⁰ growth, indicating that the TEMED-originated SEI is mechanically strong to suppress the dendritic Li⁰ upon its crystallization[11].

### Electrochemical performance of TEMED-treated Li

To evaluate the electrochemical performance of the TEMED-based lithiophilic interphase, symmetric cells with pristine and TEMED-treated Li⁰ were cycled at various current densities (0.5 and 1 mA cm⁻²) with the platting/stripping capacity of 1 mAh cm⁻² in an electrolyte consisting of 1.0 M LiTFSI in 1,3-dioxolane/1,2-dimethoxyethane

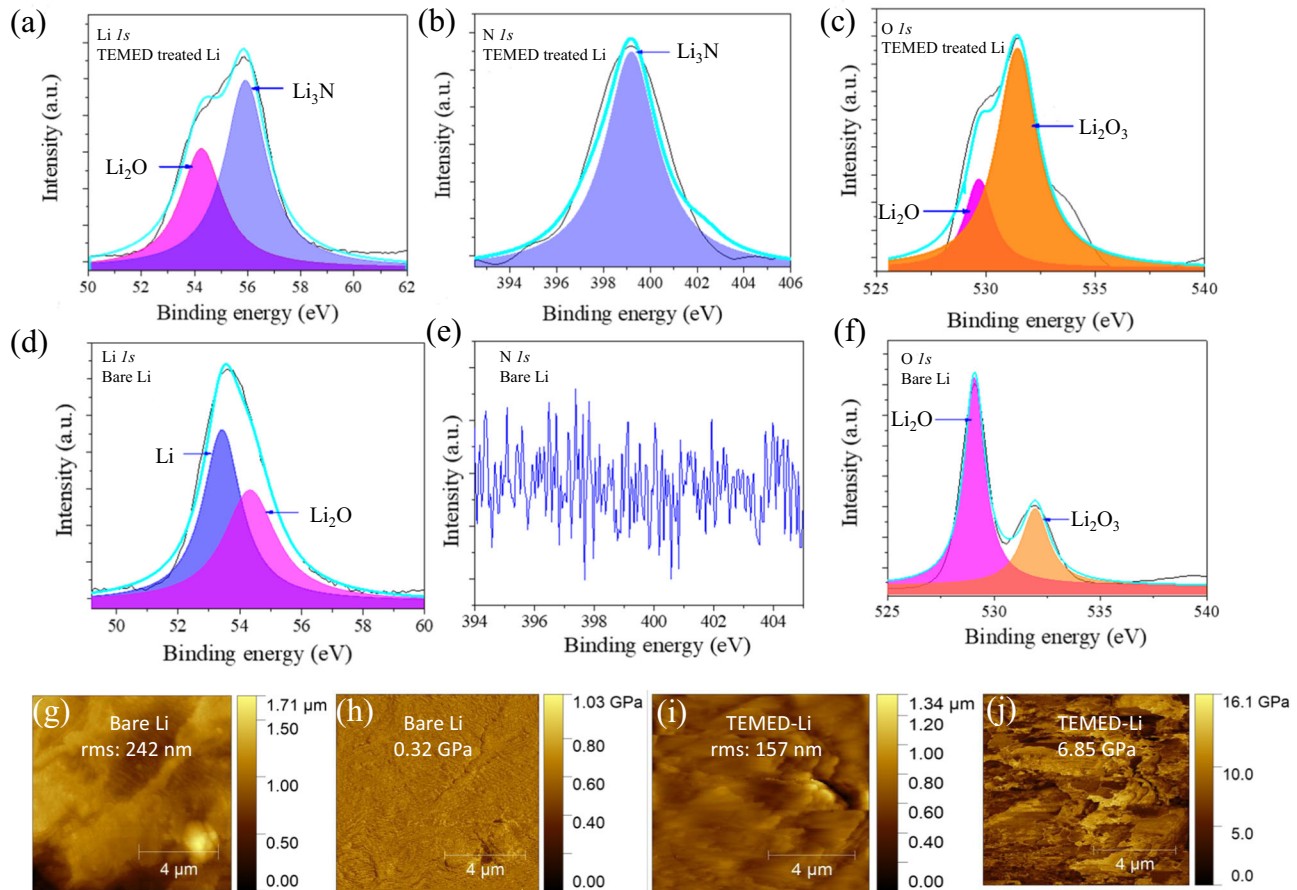

**Fig. 4 | Surface analysis of untreated Li⁰ and TEMED-treated Li⁰. a** Li *1s*, **b** N *1s*, and **c** O *1s* XPS spectra of TEMED-treated Li⁰. **d** Li *1s*, **e** N *1s*, and **f** O *1s* XPS spectra of untreated Li⁰. **g** AFM topography and **h** Young's modulus mapping of untreated Li⁰. **i** AFM topography and **j** Young's modulus mapping of TEMED-treated Li⁰.

(DOL:DME = 1:1 by vol). Voltage profile versus cycling time, and voltage hysteresis (estimated by calculating the average difference between the voltage of Li stripping/plating) versus cycle number are shown in Fig. 5.

The plating/stripping voltage profile of untreated and TEMED-treated Li⁰ was carried out to investigate the electrochemical stability of the TEMED-originated SEI. Figure 5a, c show the voltage profiles of plating/stripping for symmetrical cells constructed on untreated and TEMED-treated Li⁰ that achieved a capacity of 1 mAh cm⁻² at the current density of 0.5 and 1 mA cm⁻², respectively. At a low current density of 0.5 mA cm⁻², untreated Li⁰ exhibited large voltage divergence after 150 cycles, and short circuit after ~600 h. However, TEMED-treated Li⁰-based symmetric cell showed a stable voltage profile with hysteresis below 20 mV, reflecting the stable plating and stripping process for more than 3500 h. Even at a higher current density of 1 mA cm⁻², TEMED-treated Li⁰ showed stable plating and stripping for more than 500 cycles (1000 h), whereas untreated Li⁰ failed after ~400 h. TEMED-treated symmetric cells show stable performance even at a higher current density of 2 mA cm⁻² (Supplementary Figs. 8 and 9) and 5 mA cm⁻² (Supplementary Figs. 10 and 11). 700 h and ~350 h have been achieved for 2 and 5 mA cm⁻², respectively, for TEMED-treated Li⁰, compared to 150 and 50 h for untreated Li⁰. TEMED-treated Li⁰ also showed a stable plating and stripping cycle with commercial LiPF₆-based electrolyte (Supplementary Figs. 12 and 13). The cell showed a stable plating and stripping for more than 750 h at the current density of 1 mA cm⁻² for TEMED-derived SEI whereas untreated Li⁰ failed after only 270 h (Supplementary Figs. 14 and 15). The voltage hysteresis leads to the same conclusion for untreated Li⁰, an increase in voltage hysteresis was observed with increasing cycles in Fig. 5b, d. The

overpotential increases continuously, leading to early failure of the cell after only 200 cycles (~400 h). This large hysteresis implies the formation of a highly resistive and unstable interphase. The unstable SEI formed during cell operation continues to consume electrolyte to repair new SEI, accompanied by the formation of dendritic and dead Li⁰, eventually leading to the early failure of the cell[10]. Symmetrical cell performance of the TEMED-treated Li⁰ at high current and high capacity with 50-μm-thick Li⁰ have also been performed to determine the compatibility of TEMED-originated SEI. 900 h and ~600 h have been achieved for the current density of 1 mA cm⁻² and 5 mA cm⁻² at the capacity of 3 mAh cm⁻², respectively (Supplementary Figs. 16 and 17), for TEMED-treated Li⁰, compared to 300 and 120 h for untreated Li⁰. Even at the high capacity of 3 mAh cm⁻² with 50-μm-thick Li⁰, TEMED-originated SEI ensures longer cycle life.

To better understand the morphology of Li deposition on untreated and TEMED-treated Li⁰, we performed SEM examination after 5, 20, and 100 cycle. Figure 5e shows a schematic illustration of the growth of dendritic and dead Li⁰ on the untreated Li⁰ after plating and stripping cycles, where the native SEI from the reaction between Li⁰ and electrolytes are fragile, non-uniform, and unstable. Such SEI can be easily ruptured during electrode volume changes and by uneven plating/stripping.

In contrast, TEMED-treated Li⁰ (Fig. 5f) prevents side reactions of Li⁰ with the electrolyte. Figure 5g–l shows the SEM images of untreated Li and TEMED treated Li after 5th, 20th, and 100th plating at 0.5 mA cm⁻² with a capacity of 1 mAh cm⁻². For untreated Li⁰ we observed uneven Li⁰ plating and dendrite growth starting from the 5th cycle (Fig. 5g). The unregulated and unprotected surface of the untreated Li⁰ creates large protuberances generating a non-uniform

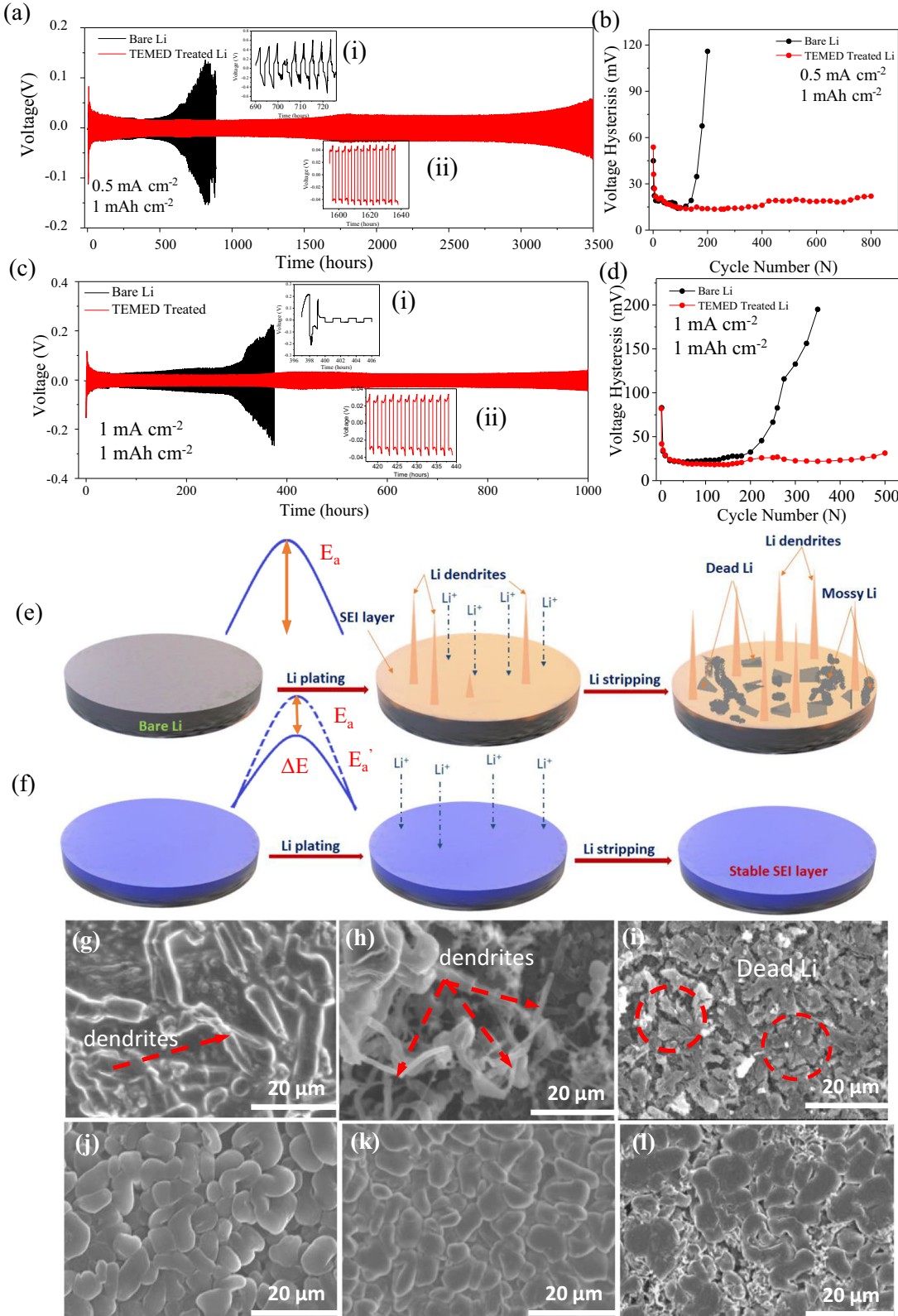

**Fig. 5 | Symmetric cell performances of untreated Li⁰ and TEMED-treated Li⁰.**
**a**, **b** Galvanostatic cycling and voltage hysteresis at 0.5 mA cm⁻²/1 mAh cm⁻². (insets show (i) short circuit for untreated Li⁰, (ii) plating/stripping behavior of TEMED-treated Li⁰ at 1600–1640 h). **c**, **d** Galvanostatic cycling and voltage hysteresis at 1 mA cm⁻²/1 mAh cm⁻². (insets show (i) short circuit for untreated Li⁰, (ii) plating/stripping behavior of TEMED-treated Li⁰ at 420–440 h). Schematic illustration of **e** Li⁰ dendrite growth on untreated Li⁰ and **f** uniform deposition on TEMED-treated artificial SEI. SEM images of **g**–**i** untreated Li⁰, **j**–**l** TEMED-treated Li⁰ at 5th, 20th, and 100th plating, respectively. The thickness of lithium chip is 450 μm. The scale bars are at 20 μm.

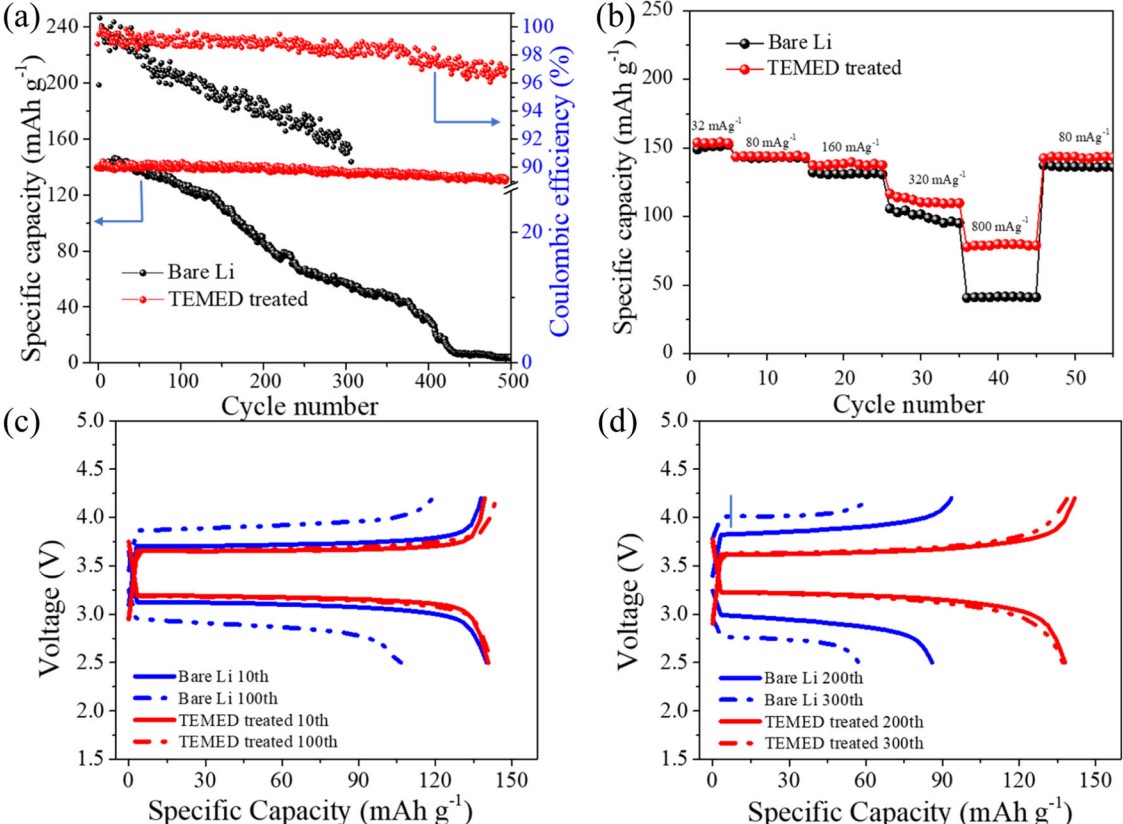

**Fig. 6 | Electrochemical performances of full cells. a** Cycling performance of full cells based on LFP positive electrode and untreated or TEMED-treated Li negative electrode at a specific current of 160 mA g⁻¹. **b** Rate performances of full cells based on LFP positive electrode and untreated or TEMED-treated Li negative electrode.

**c, d** Charge/discharge voltage profiles at different cycles of full cells based on LFP positive electrode and untreated or TEMED-treated Li negative electrode at 160 mA g⁻¹. The mass loading of LFP is -2.0 mg cm⁻².

electric field. Dendrite growth is also promoted because of the uneven surface and locally concentrated Li⁺ flux because the sharp end of these dendrites serves as a center at which charges tend to accumulate[10,51]. This needle-like structure with a sharp end may also penetrate through separator to cause an internal short circuit that results in safety issues[52,53]. In addition, the high surface area associated with the dendritic morphology and side reactions results in an extremely low CE.

In contrast, the TEMED-treated Li⁰ leads to a dense and nodule-like morphology in the absence of Li⁰ dendrites. Cross-sectional SEM (Supplementary Fig. 18) showed -50 μm thickness Li⁰ after 100 cycles as compared to -19 μm for the TEMED-treated Li⁰. Even after 100 cycles, the surface of TEMED-treated Li⁰ still maintained a compact surface without discernible dendrites (Fig. 5l). Hence, our structural characterization at a high stripping/plating rate over long cycling times further supports the hypothesis that high transference number of Li⁺ across TEMED-originated SEI improves Li⁺ mobility, which in turn decreases the concentration gradient, leading to uniformity of Li⁰ electrodeposition with suppressed lithium dendrite formation.

Nucleation overpotential is defined as the voltage difference between the beginning voltage dip and the following flat plateau during plating, which is also known as the Li nucleation barrier (Supplementary Fig. 19). Lower nucleation overpotential signifies higher lithiophilicity and is preferred for higher reversibility of Li⁰ chemistry. Untreated Li⁰ shows a higher nucleation overpotential of 37 mV indicating a significantly large energy barrier whereas, TEMED-treated Li⁰ shows higher lithiophilicity with the lowest nucleation overpotential of 15 mV. Electrochemical performance of L⁻²i⁰||Cu half cell and TEMED-Li⁰||Cu cell with the thin Li⁰ chip of 50 μm thickness have been

performed at the cycling current density of 1 mA cm⁻² and capacity of 1 and 3 mAh cm⁻² (Supplementary Fig. 20 and 21, respectively). The CE of the untreated-Cu cell decreases rapidly after only 75 cycles; however, the CE of the TEMED-Li⁰||Cu cell maintains a stable cycle during 175 cycles for a capacity 1 mAh cm at current density of 1 mA cm⁻². The enhanced CE and lifespan were also significant when the capacity was increased to 3.0 mAh cm⁻² (Supplementary Fig. 20).

To evaluate the compatibility of TEMED-treated Li⁰ as a negative electrode for practical LMBs, we adopted lithium iron phosphate (LFP) and NMC-111 as two positive electrode materials to assemble a full cell LMB. Figure 6a shows the cycling performance of the full cell using untreated or TEMED-treated Li⁰ as the negative electrode at a constant specific current of 160 mA g⁻¹. We observed linear degradation in capacity for the full cell with untreated Li⁰ as the negative electrode (Fig. 6a). However, with the TEMED-treated Li⁰, we obtained a steady and stable capacity for the full cell. In rate capability tests (Fig. 6b), at lower specific currents of 32, 80, and 160 mA g⁻¹ we observed comparable capacity for untreated and TEMED-treated Li⁰. However, high specific currents of 800 to 80 mA g⁻¹ lead to a large capacity loss for untreated Li (Supplementary Figs. 22 and 23). TEMED-treated Li⁰, on the other hand, recovers almost 100% of capacity, due to the stable dendrite-free Li⁰ plating/stripping of the TEMED-treated Li⁰. LFP-based full cell with a high mass loading (-9.5 mg cm⁻²) has also shown an improved performance for the TEMED-treated Li⁰ as compared to the untreated Li⁰. During the 10th cycle Li/LFP showed a discharge capacity of -135 mAh g⁻¹ whereas TEMED-Li⁰/LFP showed a discharge capacity -123 mAh g⁻¹ (Supplementary Fig. 24). After the 50th cycle (Supplementary Fig. 25) capacity retention of -74% has been obtained for TEMED-Li/LFP in comparison with -50% for untreated Li⁰/LFP full

cell. Similar performance has been obtained for NMC-based cells where a drastic decline in capacity for untreated Li/NMC with capacity retention of ~48% has been obtained as compared to 73% for TEMED-Li/NMC after 200 cycles (Supplementary Figs. 26–28) suggesting that the artificial SEI originated from TEMED is more stabilized.

Figure 6c, d shows the cycling performance and voltage profile of full cells using both untreated Li and TEMED-treated Li as a negative electrode. Untreated $Li^0$/LFP full cells showed a lower CE in the first cycle, which could be attributed to $Li^0$ consumption and electrolyte decomposition to form the SEI. TEMED-treated $Li^0$ showed ~100% capacity retention from the 10th discharge to the 100th discharge. In contrast, the 10th specific discharge capacity of untreated Li/LFP full cell shows a sharp decrease from ~140 to 102 mAh $g^{-1}$, retaining 72.8% at the 100th cycle. With increased cycle numbers, the decrease in capacity retention becomes more prominent for untreated Li/LFP. Reduced overpotentials were also observed for the TEMED Li/LFP full cell whereas, bare Li/LFP shows higher overpotential due to the loss of Li and consumption of electrolyte from side reactions and an un-stabilized SEI with dendrites, leading to overpotential and sluggish Li-ion transportation. The stable cycle life and low polarization potential suggest that the TEMED-treated Li negative electrode is capable of working under practical cycling conditions.

## Discussion

In this work, we demonstrated a facile and efficient solution processed method to provide phase pure lithium nitride ($Li_3N$) as a protective SEI, which successfully suppresses the dangerous and unstable morphologies of dendritic and dead $Li^0$, due to the low electronic conductivity and the intrinsic electrochemical stability of $Li_3N$. This artificial SEI layer offers excellent $Li^+$ conductivity with lower $Li^+$ migration energy barriers that further benefit ion transport at the interface between the electrode and electrolyte.

As a result, the TEMED-originated SEI ensures long stable plating/stripping cycling up to 3500 h at 0.5 mA $cm^{-2}$ along with a full cell cycling up to 500 cycles at 160 mAg$^{-1}$ (1C rate). These dendrite-free TEMED treated Li should facilitate applications of high energy density Li metal batteries.

## Methods

### Materials and synthesis

Li chips (diameter size = 15.6 mm and thickness = 450 μm) were purchased from MTI Corp. Tetramethylethylenediamine (TEMED) was purchased from Sigma-Aldrich. TEMED was used without any further modifications. Lithium chips were allowed to be completely immersed into the TEMED in the petri dish and were kept overnight. The Li chips were allowed to be dried at 60 °C for half an hour to let the unreacted liquid evaporate away. The dried Li chips were then used for further analysis and cell fabrication.

### Electrode fabrication

Lithium iron phosphate (LFP) powders were mixed with Super-P carbon black and polyvinylidene fluoride (PVDF) at a weight ratio of 80:10:10, respectively, in the N-methyl-2-pyrrolidone (NMP) solvent to form a slurry using the mortar and pestle. Similarly, for Lithium nickel manganese cobalt oxides (NMC) based positive electrode, NMC powders were mixed with Super-P carbon black and polyvinylidene fluoride (PVDF) at a weight ratio of 80:10:10, respectively, in the N-methyl-2-pyrrolidone (NMP) solvent to form a slurry using the mortar and pestle. The slurry was coated on an aluminum foil current collector by doctor blading and then dried in the vacuum oven at 80 °C for 12 h. The dried samples were cut into circular disks with a diameter of 12 mm and used as the working electrode. The total areal mass loading of the NMC electrode was ~2.5 mg $cm^{-2}$ and the areal mass loading of active material LFP was ~2.0 mg $cm^{-2}$. For the high mass loading full cell, the total mass loading was ~9.5 mg $cm^{-2}$.

### Electrochemical characterization

The CR-2032 Li-ion coin cell was assembled inside an argon-filled glove box (moisture and $O_2$ level <0.1 ppm) for all the electrochemical measurements. Celgard 2500 with a film thickness of 25 μm was used as a separator. The electrolyte was 1 M Lithium bis(tri-fluoromethanesulfonyl)imide (LiTFSI, Sigma-Aldrich) in 1,3-dioxolane (DOL, Sigma-Aldrich)/1, 2-dimethoxyethane (DME) Sigma-Aldrich) where the volume of both DOL and DME were 1:1 (1:1 volume ratio) with 1 wt% Li nitrate ($LiNO_3$, Alfa Aesar). Commercially available electrolyte 1 M Lithium hexafluorophosphate ($LiPf_6$) in ethylene carbonate (EC)/diethyl carbonate (DEC) (1:1 volume ratio) has also been used for the comparative analysis of the performance of the TEMED treated $Li^0$ in both the electrolytes. For the full cell, LiTFSI in DOL/DME (1% $LiNO_3$) has been used for both LFP and NMC positive electrodes. The amount of electrolyte used was controlled as ~60 μL for each cell. Cells were tested under a different current density of 0.5 and 1 mA $cm^{-2}$ with a capacity of 1 mAh $cm^{-2}$ using Land battery analyzers (CT2001A).

Electrochemical impedance spectroscopy (EIS) was carried out by a Biologic VSP potentiostat with 10 mV amplitude AC signal with frequency ranging from 0.1 Hz to 100 K Hz with 6 points per decade. The calculations of diffusion coefficient and activation energy can be found in Supplementary Notes 2 and 3. XRD of the samples was conducted on a Rigaku SmartLab diffractometer with Cu Kα radiation (λ = 1.54178 Å). Topography and Young's Modulus of untreated Li and graphite–$SiO_2$ Li were measured using an Agilent SPM 5500 atomic force microscope equipped with a MAC III controller using a tip (product RTESPA-525) with the resonance frequency of 75 kHz. Raman spectroscopy was carried out using the Horiba Raman system with a 532 nm laser.

Galvanostatic charge-discharge measurements of the coin cells were carried out using the LAND CT2001A system. Plating/stripping of the symmetric cells was performed at various areal current densities from 0.5 to 5 mA $cm^{-2}$ to achieve various areal capacities from 1 to 3 mAh $cm^{-2}$. Full cells were cycled at a constant current density of 160 mAg$^{-1}$ (1 C) and at various current density rates from 80, 160, 320, and 800 mAg$^{-1}$ for every 10 cycles and followed back to 80 mAg$^{-1}$. LFP-based full cells were cycled at the voltage range between 2.5 and 4.2 V at 160 mAg$^{-1}$ (1C) and NMC-based cells were cycled at the voltage range of 2.7 to 4.2 V. All electrochemical tests were performed at 25 °C in an environmental chamber.

SEM characterization was carried out using a Hitachi S-4300N SEM. TEM characterization was carried out using a JEOL 2100F TEM with 200 kV field emission. The samples utilized for TEM characterization were prepared by plating lithium onto a carbon film-supported copper grid by a coin cell, which served as the TEM sample support. Subsequently, these samples underwent TEMED treatment to replicate our experimental conditions.

### Phase-field simulation

The phase field simulations were performed on COMSOL. The Multi-physics software used general PDE and the solver was set as time-dependent. The details are described in Supplementary Note 4. The size of the model is chosen to be 500 × 500 μm². Dirichlet boundary conditions were selected for the Nernst-Planck equation (Equation 6 in Supplementary Information) and the current continuity equation (Equation 7 in Supplementary Information), while the zero-flux boundary condition is set for the phase-field variable ($\xi$). $C_{Li}$ is fixed at 1.0 mol/L in the electrolyte and 0.0 mol/L in the electrode, while $\phi$ is fixed at −0.35 V in the electrode and 0.0 V in the electrolyte as the boundary conditions. The initial state is a pure electrolyte, in which $\xi$ and $\phi$ are set to be zero, while the initial value for $C_{Li}$ is set to be 1.0 mol/L. Then, a semi-circle type random noise is added on the surface of the Li negative electrode, which acts as a nucleus for Li dendrite growth. The diffusivity of Li-ion is set to be $D^s = 3.05 \times 10^{-10}$ m²/s in the electrolyte solution, and $D^e = 3.1 \times 10^{-13}$ m²/s in the Li electrode, based on the activation energy of untreated Li from experimental

measurement. To study the effect of the $Li_3N$ protective layer on the Li deposition morphology, we introduced a highly diffusive SEI layer on the surface of the Li negative electrode to mimic the treated Li. The diffusivity of this layer ($D^i$) after $N_2$ and TEMED treatment increases by 2 times and 10 times than untreated Li, based on the activation energy of treated Li from experimental measurement.

## Density functional theory calculation
All density functional theory (DFT) calculations were carried out by using the Vienna ab initio simulation package (VASP)[54]. The projector-augmented wave method was used to account for core-valence interactions[55], and the generalized gradient approximation (GGA) in the form of the Perdew–Burke–Ernzerhof functional was used for the exchange-correlation interactions[56]. The electronic wave functions were represented by a plane-wave basis set with a cut-off energy of 500 eV. A $3 \times 3 \times 3$ $Li_3N$ supercell structure including 108 atoms was used, and a $2 \times 2 \times 2$ Monkhorst–Pack grid was used for the Brillouin zone integration. The convergence criteria were $1 \times 10^{-5}$ eV energy differences to solve the electronic wave function, and all atoms were relaxed until the forces were less than 0.01 eV/Å. To determine the Li diffusion pathways and migration barriers, a supercell with one Li vacancy was prepared by removing an individual Li atom. The energy profiles along Li migration were calculated as shown in Supplementary Table 1 and Supplementary Figs. 29 and 30 by the climbing-image nudged elastic band (NEB) method[57].

## Data availability
The authors declare that all the relevant data are available within the paper and its Supplementary Information file or from the corresponding authors upon request.

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

## Acknowledgements

This work is supported by the National Science Foundation under awards CBET-2312247, CBET-2038083, and OIA-2132021. A.C. and K.X. thank the support from Joint Center of Energy Storage Research, an Energy Hub funded by US Department of Energy Basic Energy Science. We also acknowledge the part support from NSF ECCS- 2240507 and Department of Defense Batteries and Energy to Advance Commercialization and National Security (BEACONS) center.

## Author contributions

J.P. and A.G. designed the experiments. A.C. performed XPS characterization. B.P. and Y.C. performed phase field simulation and analysis. W. H., A.B., Z.Y. and B.S.L. assisted in material characterization and participated in performing ex situ SEM experiments. M.Y.Y. conducted DFT calculations. S.G. reviewed the paper. X.X., Y.C., W.A.G., K.X. and Y.Z. analyzed the overall results and supervised the work. J.P. wrote the paper with assistance from coauthors. All authors have discussed the paper.

## Competing interests

The authors declare no competing interests.
