## [Peer Review File · Nature Communications]

Manipulating the Diffusion Energy Barrier at the Lithium Metal Electrolyte Interface for Dendrite-free Long-life BatteriesReviewer #1 (Remarks to the Author):

In this work, the authors designed a Li₃N-rich SEI on Li metal anode for rechargeable Li metal batteries by an immersion method. The properties of the artificial SEI and the modified Li metal anode were investigated. The performance of Li metal anode was improved under the protection of the artificial SEI. Large revision is necessary to meet the high level of the journal. The following are some tips.

- (1) Is LiTFSI-DOL/DME ether based liquid electrolyte suitable for cells when the voltage is above 4 V?
- (2) Fig. 5a and c should be enlarged to show the phenomenon of "short circuit".
- (3) Many details are missing, such as the mass loading of LFP cathode, the electrolyte used for NMC-Li coin cell. Please enrich these information.
- (4) Many careless mistakes should be avoided, such as "f Li₃N", "O₂", "LiNO₃".
- (5) Some sentences should be polished to improve the fluency.

Reviewer #2 (Remarks to the Author):

Pokharel et al reported a Li₃N-rich SEI layer derived from the TEMED on the metallic Li surface via the solution-based process. The Li electrode employed with the generated phase-pure single crystalline Li₃N displayed lower Li⁺ diffusion barrier and improved electrochemical performances, as revealed by simulations and experiments. Although the method of manipulating the diffusion barriers and kinetics of Li⁺ is very promising and interesting, there exist some critical problems and some contradictory data in the manuscript. I think the work can be published after addressing the following comments carefully.

Comments:

1. it is not persuasive to conclude that the formed Li₃N is phase-pure and single crystalline through the weak even disappeared XRD signals. How does the author rule out the nanosize effect of the Li₃N? More specific and detailed characterizations should be provided to demonstrate the presence and formation of single crystalline. Meanwhile, the Li₂O and Li_xO_y are obviously observed according to the XPS results, and therefore the enhanced performances might not be simply attributed to the presence of phase-pure Li₃N.
2. The statement that the single crystalline Li₃N can decrease Li diffusion barrier in comparison to the polycrystalline one needs to be further verified. A series of comparisons from phase field simulations to electrochemical measurements are suggested to conduct before drawing this conclusion.
3. In the cells, the Li diffusion along which direction will be accelerated? Vertical? If so, how can the Li dendrite growth be prevented and what is the mechanism for the improved spatial homogeneity? The authors are recommended to consider more about the lateral diffusion that benefits the uniform Li ion flux spreading and distribution.
4. During the immersion process, organic moieties derived from TEMED are expected to be introduced and left on the Li surface, as no further cleaning procedure is performed. The TOF-SIMS and in-depth XPS measurements are suggested to carry out to rule out the presence of organic moieties. Or, maybe the organic residues have a positive effect on the lithium plating.
5. For practical application, the areal capacities of electrodes are usually above 3.0 mAh cm⁻². Therefore, the stripping/plating capacity of 1.0 mAh cm⁻² is far behind that for practical application and a capacity above 3.0 mAh cm⁻² should be performed. The pouch cells based on large areal loading cathodes are also suggested to strengthen the argument.
6. The Coulombic efficiency tests of the Li electrodes should be given under low and high stripping/plating capacity. Also, A much thinner Li chip of 50 μm thickness is recommended to adopt in Li-Li cell and Li-Cu cells to demonstrate the superiorities.
7. In figures 5a, 5c and S6-S9, it is very confusing that the TEMED treated Li electrodes possess the much higher overpotentials than the pristine Li electrode in the initial tens of cycles while it is opposite in Figure S14. In Figure 6a, the bare Li one shows better capacities than the treated one in the first 50 cycles while it is lower than TEMED treated one in rate performance (Figure 6b), why?
8. What is the R² value in Fig S2? The error of fitting line is too large to accept. Please reperform

the measurement. There are also many confusing descriptions lacking of details, such as the electrolyte selection (with LiNO₃? Without LiNO₃?); the electrode fabrication of LFP, NCM in experimental section, and so on. What is purpose of using LTO in the manuscript? What is the immersion time used for treated Li electrode in successive tests?

9. Some recently published works on decreasing Li diffusion barriers by alloys or SACs in the lithium metal battery are suggested to be discussed in the introduction parts.

Reviewer #3 (Remarks to the Author):

In this manuscript, phase-pure/single crystalline Li₃N-rich SEI was constructed via a convenient TEMED dipping treatment. The Li₃N-rich SEI exhibited low diffusion energy barrier, high Li⁺ transference number, and outstanding mechanical to achieve dendrite-free Li plating/stripping. The TEMED treated symmetrical cell shows outstanding plating/stripping cycles with reduced overpotential and the full cell exhibits remarkably improved cycling stability at high rates compared to bare Li. The results are interesting. I would recommend it for further consideration. Some suggestions for the authors are below.

(1) For the EIS analysis of symmetric cells treated with TEMED, the impedances of charge transfer in the Figure 2f (~400 Ohm) and Figure 2d (~200 Ohm) differ significantly. Please explain the reasons for this difference.

(2) Figure 1 show that the surface morphology of the top artificial SEI appears rough and mossy accompanied with some cracks, which is ascribed to the formation of Li₃N-rich layer. Normally, the formation of cracks in SEI is not conducive to uniform the distribution of Li ions flux and the deposition of lithium.

(3) Did the content and crystal phase of Li₃N in SEI change during cycling?

(4) Please give a comparison of the performance of the Li₃N-rich SEI with other artificial layers, especially Li₃N protective layers formed by other methods.

(5) The O 1s XPS peak attributed to Li₂O is at ~531.5 eV in TEMED treated Li, while for bare Li sample, the value is ~529.5 eV. The authors do not explain the reasons for such a big difference in binding energy. I strongly suggest the authors to adjust the X-axis of the two samples and re-analyze the high-resolution deconvoluted peaks.

Thank you for your consideration of our manuscript entitled “**Manipulating Diffusion Energy Barrier at the Interface for Dendrite-free and Long-life Lithium Metal Batteries**” (NCOMMS-22-30813). We are very grateful to the three reviewers; the comments expressed by the reviewers have been carefully considered, and all necessary additions/revisions have been made based on their suggestions or criticisms in the revised manuscript.

The following are our point-by-point responses to the reviewers’ comments.

Reviewer(s)’ Comments to Authors:

Reviewer #1:

In this work, the authors designed a Li₃N-rich SEI on Li metal anode for rechargeable Li metal batteries by an immersion method. The properties of the artificial SEI and the modified Li metal anode were investigated. The performance of Li metal anode was improved under the protection of the artificial SEI. Large revision is necessary to meet the high level of the journal. The following are some tips.

Authors’ Response:

We appreciate the valuable and insightful comments on our study. The manuscript has been extensively revised and significantly improved by including additional experimental results and discussions in response to the reviewer’s comments.

(1) Is LiTFSI-DOL/DME ether based liquid electrolyte suitable for cells when the voltage is above 4 V?

Authors’ Response:

We would like to appreciate the reviewer’s valuable comment. In practical application the electrolyte performance depends on several factors, including the specific electrode materials being used, the operating conditions of the cell as temperature and current density, and the desired performance characteristics of the cell such as energy density and cycle life. However, in general LiTFSI/DOL-DME ether-based liquid electrolyte is not suitable for operation above 4 V due to

the potential for ether decomposition or instability of the electrolyte. To address this, the addition of lithium nitrate (LiNO_3), fluoroethylene carbonate (FEC) improves the stability of the LiTFSI/DOL-DME electrolyte at high voltages by reducing the formation of gas and the degradation of the electrode/electrolyte interface.

(2) Fig. 5a and c should be enlarged to show the phenomenon of “short circuit”.

Authors' Response:

We would like to thank the reviewer for carefully reviewing our manuscript and providing the valuable suggestion. We have inserted the inset of the respective “short circuit” phenomenon in the related figures as R1 and R2. We have updated the figures in our manuscript accordingly.

Fig R1: Long term cycling of symmetrical cells at the current density of 0.5 mA cm^{-2} (inset shows (i) short circuit for bare lithium, (ii) plating/stripping of TEMED at 1600-1640 hours.

Fig R2: Long term cycling of symmetrical cells at the current density of 1 mA cm^{-2} (inset shows (i) short circuit for bare lithium, (ii) plating/stripping of TEMED at 420-440 hours.

(3) Many details are missing, such as the mass loading of LFP cathode, the electrolyte used for NMC-Li coin cell. Please enrich these information.

Authors' Response:

We appreciate the reviewer for pointing out the missing content in our manuscript. We have made the necessary corrections and have provided the information of the mass loading for LFP ($\approx 2.0 \text{ mg cm}^{-2}$) cathode and the electrolyte that had been used for the NMC (LiTFSI in DOL/DME with 1% LiNO₃) based full cell. We have enriched that information in the revised manuscript.

(4) Many careless mistakes should be avoided, such as “f Li₃N”, “O₂”, “LiNO₃”.

Authors' Response:

We appreciate the reviewer for pointing out the typos for the elements. We have made all the necessary corrections in the revised manuscript.

(5) Some sentences should be polished to improve the fluency.

Authors' Response:

We would like to thank the reviewer for pointing out the fluency of the sentences, we have polished all sentences carefully and mistakes have been corrected accordingly.

Reviewer #2: (Remarks to the Author):

Pokharel et al reported a Li₃N-rich SEI layer derived from the TEMED on the metallic Li surface via the solution-based process. The Li electrode employed with the generated phase-pure single crystalline Li₃N displayed lower Li⁺ diffusion barrier and improved electrochemical performances, as revealed by simulations and experiments. Although the method of manipulating the diffusion barriers and kinetics of Li⁺ is very promising and interesting, there exist some critical problems and some contradictory data in the manuscript. I think the work can be published after addressing the following comments carefully.

Response to the reviewer: We really appreciate the positive comments on our work. The manuscript has been extensively revised and significantly improved by including additional experimental results and discussions in response to the reviewer's comments.

1. It is not persuasive to conclude that the formed Li_3N is phase-pure and single crystalline through the weak even disappeared XRD signals. How does the author rule out the nanosize effect of the Li_3N ? More specific and detailed characterizations should be provided to demonstrate the presence and formation of single crystalline. Meanwhile, the Li_2O and LiN_xO_y are obviously observed according to the XPS results, and therefore the enhanced performances might not be simply attributed to the presence of phase-pure Li_3N .

Authors' Response:

We agree with the reviewer regarding the presence of the weak XRD peak. Based on the suggestions, we reformed the XRD tests with a higher resolution. The resulting XRD signals exhibit distinct pure (001) and (002) Li_3N peaks as shown in Fig R3. The huge difference can be found compared to polycrystal Li_3N shown below.

Fig R3: XRD patterns of (a) TEMED treated Li metal and (b) polycrystal Li_3N electrode with conventional method.

To clarify the controversial part of the Li_2O and LiN_xO_y in our XPS data, an in-depth XPS measurement has been performed. In Fig R4, the XPS data of the N1s spectra with the etching time of 30 minutes show the formation of the Li_3N , whereas the Li_2O can be observed in Li 1s spectra. The presence of the oxides in Li 1s and O 1s in the in-depth XPS results suggest that the oxides are formed due to the exposure of the sample during the analysis process.

Various literature have reported that the metallic Li reacts readily with the trace amount of residual gases, resulting in the formation of surface contamination layers primarily composed of Li_2CO_3 , Li_2O and LiOH . According to literature, XPS spectra obtained without continuous sputtering exhibits notably significantly higher levels of surface contamination. Listed below are some papers with related discussion included. Even with continuous sputter-cleaning, minor presence of Li_2O is still detected on the surface. It is noted that the whole synthesis and cell assembly were performed in a glovebox with $<1\text{ppm O}_2$ and H_2O . And all experiments were repeated many times. We do not think the superior performance of treated samples is due to the negligible presence of Li_2O and LiN_xO_y .

1. Wood, et al. *ACS Appl. Energy Mater.* 2018, 1, 9, 4493–4504
2. Jeffrey, et.al. *ACS Appl. Mater. Interfaces* 2020, 12, 41, 46015–46026
3. Hong, et.al. *Electrochimica acta* 50.2-3 (2004): 535-539.

We have updated the figure and revised the expressions accordingly in the revised manuscript. It should be emphasized that the simulation models are also based on this crystal structure and the calculated results are consistent with our experimental data, as shown in the revised manuscript.

Fig R4: (a)XPS L1s spectra for TEMED treated Li (b) N1s spectra for TEMED treated Li (c) O1s spectra for TEMED treated Li and (d)XPS L1s spectra Bare Li (e) N1s spectra for Bare Li and (f).O1s spectra for Bare Li.

2. The statement that the single crystalline Li_3N can decrease Li diffusion barrier in comparison to the polycrystalline one needs to be further verified. A series of comparisons from phase field simulations to electrochemical measurements are suggested to conduct before drawing this conclusion.

Authors' Response:

We appreciate the reviewer for raising the valuable comment to perform the comparative analysis between the single crystalline and polycrystalline Li_3N .

To prove our hypothesis that the single crystalline Li_3N can decrease the Li diffusion barrier, EIS tests were conducted, and the Arrhenius plots are shown in Fig R5. The dependence of diffusion coefficient D on temperature T can be approximately described by an Arrhenius relationship. The relationship between the $\ln D_{\text{Li}}$ and $1/T$ is linear and the activation energy can be obtained from the slope of the fitting line. The results show that the bare Li, N_2 -treated Li and TEMED-treated Li have activation energy of 0.723 eV, 0.613 eV, and 0.480 eV, respectively

We further performed the integrated phase field simulations to investigate the activation energy of Li^+ ion and its transport behavior at the interface on bare Li anodes, N_2 treated Li and TEMED treated Li anodes with Li_3N as an artificial solid-electrolyte interface (SEI) layer (Fig 6 a-c). The observations showed that on bare lithium, Li dendrites grew into filament-like structures with side branches, and on N_2 treated lithium, dendrite growth was slower and the side growth was minimal. In contrast, on the TEMED treated lithium with the Li_3N layer, the initial protrusion exhibited a dome-like morphology with a smooth electrode-electrolyte interface, and its growth rate was significantly reduced. This indicated that the artificial SEI layer with higher Li-ion diffusivity and transference number could suppress Li dendrite growth from the anode.

We further analyzed the Li-ion concentration across the dendrite tip and were able to find that Li-ion concentration increased sharply for bare Li anodes, while it increased gradually for treated Li anodes, supporting the notion that higher Li-ion concentration gradients at the dendrite tip facilitate dendrite growth (Fig 6 d-f). Further analyzing the electric field variation in the different Li anodes, electric field was maximum for bare Li at the tip as compared to N_2 treated and TEMED treated Li anodes. This was attributed to the sharper tip morphology and larger curvature, which led to a higher Li-ion concentration gradient and facilitated dendrite growth. In contrast, treated Li anodes exhibited less significant variations in the electric field at different time steps, indicating inhibition of Li dendrite growth.

These results suggest that the Li_3N protective layer as an artificial SEI layer can effectively suppress Li dendrite growth by enhancing Li-ion diffusivity and reducing Li-ion concentration gradients at the dendrite tip, thereby inhibiting the self-accelerating process of dendrite growth on bare Li anodes.

Fig. R5 Correlation between $\ln D$ and reciprocal temperature (Arrhenius-plot) for TEMED treated Li and Bare-Li. The ionic conductivity of Li_3N can be calculated using the equation $\sigma = 2L/Ra$, in which, L is the thickness of Li_3N , R is the resistance of Li_3N and a is the area.

We have added the detailed discussion and data in our revised manuscript.

Fig R6. Phase-field simulation of Li dendrite growth from bare Li anode, N_2 treated Li anode TEMED treated Li anode covered with Li_3N protective layer of high Li ion diffusivity. Dendrite morphology grown on bare lithium metal (a), N_2 treated Li anode (b), and TEMED treated Li anode (c) represented by phase-field variable ξ . (d-f) 1D evolutions of Li^+ concentration profile along x -axis across the tip of the dendrite for bare lithium (d), N_2 treated lithium metal (e), and TEMED treated Li metal (f). The images on the inset show the 2D map of the Li^+ concentration at $t = 400$ s. (g-i) 1D evolutions of electric field profile along x -axis across the tip of dendrite for bare lithium (g), N_2 treated lithium metal (h), and TEMED treated Li metal (i). The images on the inset shows the 2D distribution of the local electric field at $t = 400$ s.

3. In the cells, the Li diffusion along which direction will be accelerated? Vertical? If so, how can the Li dendrite growth be prevented and what is the mechanism for the improved spatial homogeneity? The authors are recommended to consider more about the lateral diffusion that benefits the uniform Li ion flux spreading and distribution.

Authors' Response:

We appreciate the reviewers' valuable comments. We agree with the reviewer that the diffusion directions are important for the uniform Li ion flux. In response to the reviewer's valuable comment, we have performed lateral and vertical diffusion calculations by using the density functional theory (DFT).

In the calculation models, we considered all possible migration pathways including lateral (or in-plane; $\perp c$ axis) and vertical (or out-of-plane; $\parallel c$ axis) diffusions (Fig. R6). For the lateral diffusion, a Li can diffuse along the Li(2)-N plane (path *i*) or the Li(1) plane (path *ii*), where the diffusion *via* path *i* shows a much lower energy barrier (0.01 eV vs. 1.0 eV). For the vertical diffusion, a Li can diffuse between the Li(2)-N plane directly (paths *iii* and *iv*) or passing through the Li(1) plane (path *v*). Paths *iii* and *iv* show ~ 0.6 eV of energy barrier, whereas path *v* shows 1.8 eV. Because of the lowest barrier, the lateral diffusion *via* path *i* is most likely dominant for Li diffusion in a-Li₃N. Also, the results indicate that the Li diffusion passing through the Li(1) plane has a high barrier for diffusion, and thus Li in the Li(2)-N plane is responsible for the most diffusion. Hence, the lateral diffusion dominates the Li ion diffusion at the interface, which can largely benefit the uniform deposition of Li metal.

Fig. R7. The estimated energy profiles for the Li migration in a- Li_3N ; (a) path *i*, (b) path *ii*, (c) path *iii*, (d) path *iv*, and (e) path *v*, respectively (each pathway is indicated in Fig. x1). Only one step migration was considered for path *i* consisting of equivalent steps, and the half step migration was calculated (and doubled for plotting) for path *v* consisting of two equivalent steps.

Fig. R8. (a) Top view and (b) side view of the supercell ($3 \times 3 \times 3$) structure of a- Li_3N , and possible Li migration pathways.

Path *i*

Distance (Å)	Normalized	Energy (eV)
0.000	0.000	0.000
0.185	0.208	0.004
0.371	0.417	0.011
0.543	0.610	0.010
0.716	0.804	0.004
0.890	0.890	0.000

Path *ii*

Distance (Å)	Normalized	Energy (eV)
0.000	0.000	0.000
0.445	0.122	0.159
1.104	0.302	0.671
1.831	0.500	1.047
2.385	0.651	0.811
2.916	0.797	0.373
3.366	0.919	0.082
3.661	1.000	0.000

Path *iii*

Distance (Å)	Normalized	Energy (eV)
0.000	0.000	0.000
0.291	0.076	0.019
0.687	0.180	0.156
1.083	0.283	0.368
1.490	0.390	0.553
1.912	0.500	0.630
2.432	0.636	0.514
2.927	0.766	0.264
3.415	0.894	0.046
3.821	1.000	0.000

Path iv

Distance (Å)	Normalized	Energy (eV)
0.000	0.000	0.000
0.456	0.100	0.083
1.047	0.229	0.335
1.659	0.363	0.454
2.279	0.499	0.569
2.790	0.611	0.481
3.285	0.719	0.386
3.777	0.827	0.240
4.246	0.930	0.034
4.567	1.000	0.000

4. During the immersion process, organic moieties derived from TEMED are expected to be introduced and left on the Li surface, as no further cleaning procedure is performed. The TOF-SIMS and in-depth XPS measurements are suggested to carry out to rule out the presence of organic moieties. Or, maybe the organic residues have a positive effect on the lithium plating.

Authors' Response:

We appreciate the reviewer's suggestions to improve the novelty and importance of our work. In response to the reviewer's valuable comment, the in-depth XPS measurement was performed to

investigate the organic moieties. According to Fig. R9, the N1s spectra indicate the formation of Li_3N between TEMED and Li metal after a 30-minute etching time. Conversely, for the untreated Li, the N1s spectra do not exhibit any noticeable peak. The in-depth XPS analysis reveals the absence of organic compounds, but both the treated and untreated samples exhibit the presence of Li_2O . The detection of Li 1s and O 1s peaks in the in-depth XPS results suggests that the oxides are formed as a result of sample exposure during the analysis process, which is normal and also appears in the literature as we discussed in Question 1. The absence of organic compounds in the in-depth XPS indicates that the enhanced cell performance is solely due to the presence of Li_3N , as no other substances were detected.

Fig R9: (a)XPS L1s spectra for TEMED treated Li (b) N1s spectra for TEMED treated Li (c) O1s spectra for TEMED treated Li and (d)XPS L1s spectra Bare Li (e) N1s spectra for Bare Li and (f).O1s spectra for Bare Li.

5. For practical application, the areal capacities of electrodes are usually above 3.0 mAh cm^{-2} . Therefore, the stripping/plating capacity of 1.0 mAh cm^{-2} is far behind that for practical application and a capacity above 3.0 mAh cm^{-2} should be performed. The pouch cells based on large areal loading cathodes are also suggested to strengthen the argument.

Authors' Response:

We appreciate the reviewer's concern about the performance of the battery with higher current densities. We performed the symmetrical cell testing with Li with the thickness of 50 μm thickness at the higher capacity of 3 mAh cm^{-2} and current density of 1 mA cm^{-2} and 5 mA cm^{-2} . Full cell with 50 μm thick Li (for both TEMED treated and Bare Li) and LFP as cathode have been analyzed. The mass loading of the cathode was $\sim 9.5 \text{ mg/cm}^2$.

Fig R10 and R11 shows the voltage profile of Li plating/ stripping of bare Li and TEMED treated Li symmetrical cells that achieved a capacity of 3 mAh cm^{-2} at the current density of 1 mA cm^{-2} and 5 mA cm^{-2} respectively. 900 hours and ~ 600 hours have been achieved for 1 mA cm^{-2} and 5 mA cm^{-2} respectively for TEMED treated Li, compared to 300 and 120 hours for bare Li. Even at the high capacity of 3 mAh cm^{-2} with 50 μm thick Li TEMED treated Li shows improved performance with longer cycle life.

Fig R10: Comparative voltage profiles of TEMED treated symmetrical cells with bare Li at a current density of 1 mA cm^{-2} to achieve a capacity of 3 mAh cm^{-2} with 50 um thick Li.

Fig R11: Comparative voltage profiles of TEMED treated symmetrical cells with bare Li at a current density of 1 mA cm^{-2} to achieve a capacity of 3 mAh cm^{-2} with 50 um thick Li.

We appreciate reviewer for pointing out the importance of the research in the practical application and to perform the Full cell with high mass loading. We have been performed the Full cell test where the mass loading of the cathode material (LFP is $\sim 9.5 \text{ mg/cm}^2$) and the cell with TEMED treated Li has shown an improved and stable performance as compared to the cell

with bare Li as an anode. Due to limited resources, we will set up related equipment and perform pouch cell assembly and test in our future work.

LFP based full cell with high active mass ($\sim 9.5 \text{ mg/cm}^2$) have also shown an improved performance for the TEMED treated Li as compared to the Bare Li. During the 10th cycle Li/LFP showed the discharge capacity of $\sim 135 \text{ mAh g}^{-1}$ whereas TEMED-Li/LFP showed a discharge capacity $\sim 143 \text{ mAh g}^{-1}$ (**Fig R12**). After 50th cycle (**Fig R13**) capacity retention of $\sim 77\%$ have been obtained for TEMED-Li/LFP whereas bare-Li/LFP full cell showed a degradation in retaining the capacity ($\sim 55\%$). TEMED based full cell also showed low voltage as compared to Bare-Li for the charge/discharge. This suggests that TEMED based cell have respectable stability even at very high mass density. Related results and discussions have been added into the revised manuscript.

Fig R12: Charge/discharge voltage profiles at 10th cycle for LFP coupled with bare Li and TEMED treated Li at 140 mA g^{-1} .

Fig R13: Charge/discharge voltage profiles at 50th cycle for NMC coupled with bare Li and TEMED treated Li at 140 mA g⁻¹.

6. The Coulombic efficiency tests of the Li electrodes should be given under low and high stripping/plating capacity. Also, a much thinner Li chip of 50 μm thickness is recommended to adopt in Li-Li cell and Li-Cu cells to demonstrate the superiorities.

Authors' Response:

We appreciate reviewer's suggestion to perform the coulombic efficiency test of Li electrode under low and high stripping/plating capacity. In response to the reviewer's valuable comment, the bare Li-Cu cell and TEMED Li-Cu cells were tested with 50 μm thick Li. The cells were tested at the higher current and higher capacity of 1 mA cm^{-2} , 1 mAh cm^{-2} and 1 mA cm^{-2} , 3 mAh cm^{-2} , as shown in Fig R14 and R15. The TEMED treated Li-Cu cell shows an improved stability and higher CE value as compared to the bare Li for both 1 mA cm^{-2} , 1 mAh cm^{-2} and 1 mA cm^{-2} , 3 mAh cm^{-2} . The CE of the bare Li-Cu cell decreases rapidly after only 75 cycles; however, the CE of the TEMED-Li|Cu cell maintains stable cycle during 175 cycles for capacity 1 mAh cm^{-2} at current density of 1 mA cm^{-2} . The enhanced CE and lifespan were also significant when the capacity was increased to 3.0 mAh cm^{-2} . The new data has been updated in the revised manuscript.

Fig R14: Coulombic efficiency of Li-Cu cell at the current density of 1 mA cm^{-2} with the capacity of 1 mAh cm^{-2} with $50 \mu\text{m}$ lithium chip.

Fig R15: Coulombic efficiency of Li-Cu cell at the current density of 1mA cm^{-2} with the capacity of 3mAh cm^{-2} with $50\mu\text{m}$ lithium chip.

7. In figures 5a, 5c and S6-S9, it is very confusing that the TEMED treated Li electrodes possess the much higher overpotentials than the pristine Li electrode in the initial tens of cycles while it is opposite in Figure S14. In Figure 6a, the bare Li one shows better capacities than the treated one in the first 50 cycles while it is lower than TEMED treated one in rate performance (Figure 6b), why?

Authors' Response:

We appreciate the reviewer for pointing out the discrepancy between the figures. We would like to clarify that there is a mistake while naming the x-axis in Fig. S14. We have made the correction in the revised manuscript. The x-axis should be capacity rather than time as it shows the nucleation potential. For nucleation overpotential to perform the measurement we waited for

the interface to be stable while for the symmetrical cell testing the cell were directly placed for measurement. In Fig. 5a, 5c and S6-S9, we tested the cells upon assembly. The higher overpotentials at beginning can be attributed to the stabilization at the interface between the electrode and the electrolyte. After several cycles, the overpotentials drop quickly and lower compared to bare lithium. Hence, the difference in the voltage of the figures can be due to the different testing scenarios.

We would like to clarify the confusion regarding fig. 6(a) and fig. 6(b). Fig. 6(a) represents the long term cycling performance of both the bare Li and TEMED treated Li at the C-rate of 1C. While Fig. 6(b) shows the different specific capacity of the Li and TEMED treated cell at the various C-rate. For Fig. 6a, the lower capacity of the cell with TEMED treated lithium compared to the cell with bare lithium at the beginning can be attributed to the initial stabilization of the interfacial layer, which is similar to the overpotential discussion above related to Fig. 5a. For the rate performance in Fig. 6b, we initially performed the formation cycles at a very low current density to stabilize the SEI layer before carrying out tests with different rates. Hence, the capacity of the cell with TEMED treated lithium metal is higher from the beginning compared to the cell with bare lithium since the interfacial layer stabilization has been completed.

Fig. R16: Nucleation overpotential for Bare Li and TEMED treated Li

8. What is the R2 value in Fig S2? The error of fitting line is too large to accept. Please re perform the measurement. There are also many confusing descriptions lacking of details, such as the electrolyte selection (with LiNO₃? Without LiNO₃?); the electrode fabrication of LFP, NCM in experimental section, and so on. What is purpose of using LTO in the manuscript? What is the immersion time used for treated Li electrode in successive tests?

Authors' Response:

We appreciate reviewer for suggesting repeating the experiment as the fitting line was too large. We would like to add that we performed the related experiment and have updated all the related experiments accordingly based on the activation energy obtained from the Arrhenius plot. We repeated the phase field simulation after repeating the experiment for the activation energy and have updated the revised manuscript.

Fig R16: Correlation between $\ln D$ and reciprocal temperature (Arrhenius-plot) for TEMED treated Li and Bare-Li. The ionic conductivity of Li_3N can be calculated using the equation $\sigma = 2L/Ra$, in which, L is the thickness of Li_3N , R is the resistance of Li_3N and a is the area.

We apologize for the confusion regarding the use of electrolyte. We have used 1% LiNO_3 in the LiTFSI-DOL/DME based electrolyte. We have added the details regarding the same in the updated manuscript as well.

The details regarding the cathode fabrications and the optimum emersion time regarding the TEMED treatment process have been enriched with more details in the updated manuscript.

We thank the reviewer for pointing out the presence of LTO in the manuscript. We have made corrections as it should have been NMC instead of the LTO.

9. Some recently published works on decreasing Li diffusion barriers by alloys or SACs in the lithium metal battery are suggested to be discussed in the introduction parts.

Authors' Response:

Considering the suggestion of the reviewer, we incorporated the recent publications on the decreasing Li diffusion barrier by alloys or SACs in the introduction section.

1. Ma, et al. *Small* **17** (2021), 2007142
2. Yue, et al. *Advanced Functional Materials* **31**, 2008786 (2021).
3. Jiang, et al. *Chemical Engineering Journal* **414**, 128928 (2021).
4. Liu, et al. *Energy Storage Materials* **41**, 1-7 (2021)
5. Li, et al. *Energy Storage Materials* **37**, 233-242 (2021).
6. Luo, et al. *Nano Energy* **87**, 106212 (2021).
7. Xue, et al. *Nano Energy* **79**, 105481 (2021).

Reviewer #3 (Remarks to the Author):

In this manuscript, phase-pure/single crystalline Li_3N -rich SEI was constructed via a convenient TEMED dipping treatment. The Li_3N -rich SEI exhibited low diffusion energy barrier, high Li^+ transference number, and outstanding mechanical to achieve dendrite-free Li plating/stripping. The TEMED treated symmetrical cell shows outstanding plating/stripping cycles with reduced overpotential and the full cell exhibits remarkably improved cycling stability at high rates compared to bare Li. The results are interesting. I would recommend it for further consideration. Some suggestions for the authors are below.

Response to the reviewer: We greatly thank the valuable comments from the review. The manuscript has been revised extensively based on the suggestions.

(1) For the EIS analysis of symmetric cells treated with TEMED, the impedances of charge transfer in the Fig. 2f (~400 Ohm) and Fig. 2d (~200 Ohm) differ significantly. Please explain the reasons for this difference.

Authors' Response:

We appreciate reviewer for carefully reading and raising this good point.

The EIS in Fig. 2d is the initial EIS for the bare lithium and TEMED treated Li, where the charge transfer resistance is ~400 ohms and 200 ohms respectively.

Fig. 2f represents the EIS before and after the polarization for the calculation of the transference number. We performed charge and discharge cycles at 0.01 mA cm^{-2} , with 4 hour charge, 30 minute rest and 4 hour discharge, with the process repeated 6 times. The cell was then polarized at 10 mV for 10 hours to ensure a steady state (**Fig. 2 e,f**). EIS spectra before polarization and after the steady-state had been reached is shown in inset of **Fig. 2 e,f**. Hence, the testing conditions between Fig. 2f and Fig. 2d are quite different, showing a different charge transfer resistance.

(2) Fig. 1 show that the surface morphology of the top artificial SEI appears rough and mossy accompanied with some cracks, which is ascribed to the formation of Li_3N -rich layer. Normally, the formation of cracks in SEI is not conducive to uniform the distribution of Li ions flux and the deposition of lithium.

Authors' Response:

We appreciate the reviewer's valuable and insightful comment on our study.

We agree that the formation of SEI layer is a complex electrochemical process, a rough and mossy SEI layer may have a non-uniform distribution of Li ion flux due to the presence of surface irregularities and variations in the thickness of the layer.

To examine the artificial SEI layer, we performed various experiments to compare TEMED treated Li and bare Li. AFM measurements showed the average RMS roughness of 242 and 157 nm, for Bare Li and TEMED respectively. The higher RMS value for bare Li implies uneven and rough

surfaces that can create large protuberances on the electrode surface³. These protuberances generate non-uniform electric fields during charge/discharge leading to inhomogeneous plating of Li. In contrast, the smooth surface of TEMED treated Li electrode based on the AFM tests provides a route for uniform Li plating. Similarly the an average Young's modulus values of 0.32 and 6.85 GPa, for Bare Li and TEMED treated Li indicating that the TEMED treated Li electrode can withstand mechanical forces, providing the desired mechanical stability during Li plating/stripping, as well as offering high resistance and sufficient strength to suppress Li dendrite growth⁴.

In addition, the Arrhenius-plot showed that the activation energy of 0.723 eV for bare Li, N₂ treated Li of 0.613 eV whereas TEMED treated Li showed an activation energy of 0.48 eV. This decrease in activation energy leads to a much higher mobility of Li⁺, which in turn decreases the concentration gradient to provide a more uniform surface for Li⁺ migration and plating.

The SEM image after multiple cycles also supports the results from other measurements. The unregulated and unprotected surface of the bare Li creates large protuberances generating a non-uniform electric field, leading to inhomogeneous plating of Li. Whereas the TEMED treated Li leads to a dense and nodule-like morphology with no lithium dendrites observed. This compact and nodular artificial layer serves as a physical protection barrier to inhibit penetration of organic electrolyte which could subsequently corrode the underlying Li electrode.

Hence an artificial SEI achieved through TEMED-treated Li offers lower impedance with a lower energy barrier for Li⁺ ion migration, further benefiting ion transport at the interface between electrode and electrolyte. This effectively facilitates transport of Li⁺ ions across the electrode surface, leading to high transference number and excellent mechanical strength that tolerates volume change to enforce more uniform Li⁺ ion flux. TEMED treated symmetrical cell also showed an outstanding plating/ stripping cycles with reduced overpotential and the full cell exhibits remarkably improved cycling stability and capacity retention as well as capacity utilization at high rates compared to untreated bare Li.

(3) Did the content and crystal phase of Li₃N in SEI change during cycling?

Authors' Response:

We appreciate the reviewer's concern regarding the content and phase change in the SEI after cycling of the cell. We performed the XRD to examine the structural stability of Li_3N after 100 cycles. The XRD showed the distinct and strong α -phase Li_3N peaks at (22.9° (001), and 46.6° (002)) which reveal that Li_3N is remains structurally intact during charge and discharge process and still serves as a stable SEI. The related discussion has been added into the manuscript.

Fig R17: XRD patterns of TEMED treated Li metal after 100 cycles at current of 0.5 mA cm^{-2} with total capacity of 1 mAh cm^{-2} .

(4) Please give a comparison of the performance of the Li_3N -rich SEI with other artificial layers, especially Li_3N protective layers formed by other methods.

Authors' Response:

The use of a Li_3N -rich layer as a protective coating has shown promise as a solid-state ionic conductor for lithium metal batteries, primarily due to its high ionic conductivity at room temperature and high Young's modulus. Various techniques have been employed to analyze the effectiveness of the Li_3N -rich layer as a protective coating. These include direct exposure to nitrogen gas, plasma treatment of the Li metal surface, gas reactions between liquid Li and N_2 gas, and direct pressing of Li_3N onto Li metal. These results have highlighted the potential of Li_3N as an artificial SEI material for lithium metal batteries. The application of Li_3N coatings has demonstrated their ability to reduce dendrite growth and enhance battery stability. However, concerns remain regarding the long-term stability and the achievement of higher capacity performance.

In our work, the TEMED treated Li based SEI have shown exceptional results at higher capacity and longer cycle life. We demonstrated long stable plating/stripping cycling up-to 3500 hours at 0.5 mA cm^{-2} and capacity 1 mAh cm^{-2} , ~ 600 hours have been achieved for the current density of 5 mA cm^{-2} at the capacity of 3 mAh cm^{-2} for $50 \mu\text{m}$ thick Li. Similarly full cell showed exceptional performance up-to 500 cycles at 1C rate. Comparative analysis has been shown in the table below.

Comparative performance analysis of the different Li_3N -rich artificial layers as protective layers has also been incorporated in the manuscript.

Table 1. Comparative Li_3N -rich layer as an artificial protective layers.

Synthesis process	Test condition	Cycle stability	Reference
Roll press Li_3N on lithium metal	$1 \text{ mA cm}^{-2} / 2 \text{ mAh cm}^{-2}$	100 cycles	(1)
N_2 gas reaction between pure N_2 gas and molten Li metal	1C	500 cycles	(2)
Plasma activation	$1 \text{ mA cm}^{-2} / 2 \text{ mAh cm}^{-2}$	500 cycles	(3)
Li_3N film on Li surface by nitridation process	_____	Cycling efficiency of ~90% after 40 cycle	(4)
N_2 gas reaction for 1 hour at room temperature	_____	Li_3N electrode showed increased efficiency by 15% after 30 cycle compared to bare Li	(5)
Mg_3N_2 plated on Li metal to generate Li_3N rich solid electrolyte interface	$1 \text{ mA cm}^{-2} / 1 \text{ mAh cm}^{-2}$	300 hours	(6)
TEMED treated Li	$0.5 \text{ mA cm}^{-2} / 1 \text{ mAh cm}^{-2}$ $1 \text{ mA cm}^{-2} / 3 \text{ mAh cm}^{-2}$ Full cell- 1C	3500 hours 900 hours 500 cycles	Our work

1. Park K, Goodenough JB. Dendrite-suppressed lithium plating from a liquid electrolyte via wetting of Li₃N. *Advanced Energy Materials* **7**, 1700732 (2017).
2. Li Y, *et al.* Robust pinhole-free Li₃N solid electrolyte grown from molten lithium. *ACS central science* **4**, 97-104 (2018).
3. Chen K, *et al.* Flower-shaped lithium nitride as a protective layer via facile plasma activation for stable lithium metal anodes. *Energy Storage Materials* **18**, 389-396 (2019).
4. Zhang Y, *et al.* An ex-situ nitridation route to synthesize Li₃N-modified Li anodes for lithium secondary batteries. *Journal of Power Sources* **277**, 304-311 (2015).
5. Wu M, Wen Z, Liu Y, Wang X, Huang L. Electrochemical behaviors of a Li₃N modified Li metal electrode in secondary lithium batteries. *Journal of Power Sources* **196**, 8091-8097 (2011).
6. Dong Q, Hong B, Fan H, Jiang H, Zhang K, Lai Y. Inducing the formation of in situ Li₃N-rich SEI via nanocomposite plating of Mg₃N₂ with lithium enables high-performance 3D lithium-metal batteries. *ACS Applied Materials & Interfaces* **12**, 627-636 (2019).

(5) The O 1s XPS peak attributed to Li₂O is at ~531.5 eV in TEMED treated Li, while for bare Li sample, the value is ~529.5 eV. The authors do not explain the reasons for such a big difference in binding energy. I strongly suggest the authors to adjust the X-axis of the two samples and re-analyze the high-resolution deconvoluted peaks.

Authors' Response:

We appreciate the reviewer's valuable and insightful comment on our study. Various factors can cause the shift in the binding energy of which the chemical shift or binding energy shift is more prominent¹. We re-performed the in-depth XPS analysis, which helps resolve the chemical shifts. We observed that with 30 minutes of etching time, the binding energy of the Li₂O has the same binding energy for both the TEMED treated Li and Bare Li. The manuscript has been updated with the revised in-depth XPS measurement.

1. Greczynski, et al. *Journal of Applied Physics*, 132(1), 011101(2022).

Fig R18: (a)XPS L1s spectra for TEMED treated Li (b) N1s spectra for TEMED treated Li (c) O1s spectra for TEMED treated Li and (d)XPS L1s spectra Bare Li (e) N1s spectra for Bare Li and (f).O1s spectra for Bare Li.

Reviewer #2 (Remarks to the Author):

The Li⁺ diffusion kinetics is addressed by introducing Li₃N-rich SEI layer on the Li surface, showing the exciting electrochemical results. Most of the raised comments have been solved. However, some minor issues are suggested to be addressed before publishing. The specific comments are listed as following:

1. Generally, the presence of single crystalline Li₃N is proved by Single-crystal X-ray Diffraction. Otherwise, it is hard to achieve the conclusion of single crystalline. Does this single crystalline Li₃N transform into polycrystalline Li₃N when cycling for hundreds of cycles?
2. In the in-depth XPS result, the evolution of C1s spectrum in the TEMED treated Li should be given to completely rule out the presence of upper remained organic moieties in the TEMED treated Li.
3. The electrolyte described in Line 291-292 is used without LiNO₃? Does the electrolyte (LiTFSI in DOL/DME (1% LiNO₃)) remain stable under the high upper voltage of 4.2V in the NCM electrode system? And the overpotentials of NCM full cells seems larger of ~ 1.3V.
4. The detailed fabrication of NMC electrode should also be included in the experiment.
5. For better reading, the font size in the figures could be unified.

Reviewer #3 (Remarks to the Author):

Most of questions have been addressed, while there are some new comments are given as below:

1. The authors should carefully check the results, such as Fig. S21 and S22. In additions, the polarization voltage is too high, and how about the energy efficiency?
2. As presented in Fig. S18 and 19, the initial coulombic efficiency is too high in the ether electrolyte with LiNO₃ additives. More explanation should be offer. In addition, too much electrolyte was added, how about the electrochemical performance with 30uL electrolyte?

Thank you for your consideration of our manuscript entitled “**Manipulating Diffusion Energy Barrier at the Interface for Dendrite-free and Long-life Lithium Metal Batteries**” (NCOMMS-22-30813). Our gratitude extends to the reviewers whose feedback has been thoughtfully addressed. The following are our point-by-point responses to the reviewers’ comments. All essential additions and revisions suggested or critiqued from the reviewers have been incorporated in the revised manuscript.

Reviewer #2 (Remarks to the Author):

The Li^+ diffusion kinetics is addressed by introducing Li_3N -rich SEI layer on the Li surface, showing the exciting electrochemical results. Most of the raised comments have been solved. However, some minor issues are suggested to be addressed before publishing. The specific comments are listed as following:

Authors’ Response:

We appreciate the valuable and insightful comments on our study. The manuscript has been extensively revised and significantly improved by including additional experimental results and discussions in response to the reviewer’s comments.

1. Generally, the presence of single crystalline Li_3N is proved by Single-crystal X-ray Diffraction. Otherwise, it is hard to achieve the conclusion of single crystalline. Does this single crystalline Li_3N transform into polycrystalline Li_3N when cycling for hundreds of cycles?

Response: We appreciate the reviewer’s concern regarding the crystalline of Li_3N . We agree that Single-crystal X-ray Diffraction characterization can be a good way to examine the crystallinity.

Due to the availability and ease of the sample preparation, we have opted for TEM as an alternative method to verify the presence of a single crystal. This method is also widely reported to characterize single crystalline of battery materials (*Nature Energy*, 8, 340 (2023); *Energy Storage Materials*, 62, 102947 (2023)). The TEM image and its corresponding diffraction patterns shown in Fig. R1 provide supporting evidence for the formation of hexagonal α -phase single crystal Li_3N . The diffraction pattern exhibits a single crystalline behavior. The lattice distance measures 0.19 nm, matching the XPS standard card and correlating with our XRD result. Combining our previous XRD results, we conclude that the hexagonal α -phase single crystal lithium nitride has been formed on Li electrode with TEMED. The related discussion has been added in the revised manuscript.

Fig. R1: (a)TEM image and its corresponding diffraction patterns from (0 0 1) axis of TEMED treated lithium; (b)XRD patterns of TEMED treated Li metal after 100 cycles at current of 0.5 mA cm^{-2} with total capacity of 1 mAh cm^{-2} .

We performed the XRD tests to examine the structural stability of Li_3N after 100 cycles. The XRD spectra (Fig. R1) exhibits invisible change compared to the sample before cycling (Fig. 2c). The distinct and strong α -phase Li_3N peaks at (22.9° (001), and 46.6° (002)) which reveal that Li_3N is remains structurally intact during charge and discharge process and still serves as a stable SEI. This XRD result could indicate that there is no significant phase or crystal change during cycling. The related discussion has been added into the manuscript.

2. In the in-depth XPS result, the evolution of C1s spectrum in the TEMED treated Li should be given to completely rule out the presence of upper remained organic moieties in the TEMED treated Li.

Fig. R2: (a) XPS C1s spectra for Bare Li (b) C1s spectra for TEMED treated Li.

Response: We agree with the reviewer, to rule out the presence of any organic moieties, C1s spectrum should be analyzed. We performed an in-depth XPS measurement with the etching time of 30 minutes. The C1s spectrum for both the bare-Li and TEMED treated Li showed the same peak for C-C/C-H peak at the binding energy of 285 eV. The presence of same spectrum for both

bare and TEMED treated Li confirms that there is no presence of any other organic moieties, and the excellent performance of battery is solely by the presence of Li_3N .

3. The electrolyte described in Line 291-292 is used without LiNO_3 ? Does the electrolyte (LiTFSI in DOL/DME (1% LiNO_3)) remain stable under the high upper voltage of 4.2V in the NCM electrode system? And the overpotentials of NCM full cells seems larger of $\sim 1.3\text{V}$.

Response: We apologize for the confusion regarding the use of electrolyte. We have used 1% LiNO_3 in the LiTFSI-DOL/DME based electrolyte. We have added the details regarding the same in the updated manuscript as well. The stability of NMC with LiTFSI in DOL/DME (1% LiNO_3) above 4.2 V depends on various factors including the composition of NMC cathode and the concentration of LiTFSI in the electrolyte. We did find that the addition of LiNO_3 improves the stability of the LiTFSI/DOL-DME electrolyte at high voltages by reducing the formation of gas and the degradation of the electrode/electrolyte interface. NMC is considered as stable cathode, however, operating cell above 4.2V can lead to risk of electrolyte decomposition and potentially affecting the overall performance and cycle life of the cell. We also apologize for the mistake regarding Supplementary Fig. 25. The high overpotential is based on LFP rather than NCM. We have corrected the figure caption. The full cell performance based on NCM is exhibited in Supplementary Figs. 26 and 27 with normal overpotential.

4. The detailed fabrication of NMC electrode should also be included in the experiment.

Response: We appreciate the reviewer for pointing out the missing fabrication process of NMC in the manuscript. For lithium nickel manganese cobalt oxides (NMC) based cathode, NMC powders

were mixed with Super-P carbon black and polyvinylidene fluoride (PVDF) at a weight ratio of 80:10:10, respectively, in the N-methyl-2-pyrrolidone (NMP) solvent to form a slurry using the mortar and pestle. We have added the detailed process in the updated manuscript.

5. For better reading, the font size in the figures could be unified.

Response: We appreciate the reviewer's suggestions to unify the font sizes in the figures for better visibility of figures. We have made changes and have updated the figures in the revised manuscript accordingly.

Reviewer #3 (Remarks to the Author):

Most of questions have been addressed, while there are some new comments are given as below:

1. The authors should carefully check the results, such as Fig. S21 and S22. In addition, the polarization voltage is too high and how about the energy efficiency?

Response: We appreciate the reviewer's valuable and insightful comment on our study. We would like to clarify that Fig. S21 (Fig. S23 in the revised manuscript), is based on different current density for LFP based cathode. As C-rate is a measure of the rate at which a battery is charged or discharged relative to its capacity, lower C-rate results in slower charge and discharge rates allowing batteries to typically deliver their full rated capacity at these low rates. Whereas, higher C-rates indicate faster charging and discharging, resulting in a reduced rated capacity. Hence, at higher C-rate, due to increased current flow and associated higher internal resistance losses, the polarization voltage is more prominent as C-rate increases. It also depends on the cell and testing conditions. Our goal is to compare the performances of cells based on bare Li and TEMED treated Li to elucidate the strategies to stabilize the Li metal electrode. The efficiency and overpotential

have obviously been improved with TEMED treated Li as shown in Supplementary Figs. 22 and 23 in the revised manuscript.

Fig. R3: Charge/discharge voltage profiles at 50th cycle for LFP coupled with untreated Li and TEMED treated Li at 140 mA g⁻¹ with 450 μm lithium.

Regarding Fig. S22 (Fig. S24 in the revised manuscript), we used 50 μm thick Li with rate of 1C. Fig. R3 shows the Charge/discharge voltage profiles at 50th cycle of full cell based on 450 μm Li. We did find the increase of overpotential when using thinner Li, which is expected. A thicker lithium electrode has lower resistance as it provides a larger conductive pathway for the movement of lithium ions during charge and discharge resulting in lower polarization voltage (Singh, Madhav, Jörg Kaiser, and Horst Hahn. "Thick electrodes for high energy lithium ion batteries." *Journal of The Electrochemical Society* 162.7 (2015): A1196.). Whereas, thinner lithium electrode have higher

resistance due to its smaller cross-sectional area for ion transport leading to higher polarization voltage. Moreover, we will continue to optimize the variables to decrease overpotential and improve the efficiency with thin Li electrode in our future studies.

2. As presented in Fig. S18 and 19, the initial coulombic efficiency is too high in the ether electrolyte with LiNO₃ additives. More explanation should be offered. In addition, too much electrolyte was added, how about the electrochemical performance with 30 μ l electrolyte?

Response: We appreciate the reviewer's comment on the cell performance. The electrochemical performance on both the Li-Cu cells and TEMED treated Li-Cu cell were tested with 50 μ m thick lithium and with minimum electrolyte (30 μ l) as compared to the other electrochemical tests. The objective is to study the effect of TEMED treated surface on the stabilization of the Li metal electrode. In order to reduce the influence of Cu, the cells were initially applied to a very low current density (0.1 mA cm⁻²) to stabilize the electrodes and were then tested on higher current density and higher capacity of 1 mA cm⁻², 1 mAh cm⁻² and 1mA cm⁻², 3 mAh cm⁻². This is the reason a high initial CE was obtained.

Reviewer #2 (Remarks to the Author):

The authors have well addressed all the comments and I recommend to publish it in Nature Communications.